# Human biallelic MFN2 mutations induce mitochondrial dysfunction, upper body adipose hyperplasia, and suppression of leptin expression

Nuno Rocha[1,2], David A Bulger[1,2,3], Andrea Frontini[4], Hannah Titheradge[5,6], Sigrid Bjerge Gribsholt[7], Rachel Knox[1,2], Matthew Page[8], Julie Harris[1,2], Felicity Payne[9], Claire Adams[1,2], Alison Sleigh[10,11], John Crawford[12], Anette Prior Gjesing[13], Jette Bork-Jensen[13], Oluf Pedersen[13], Inês Barroso[9], Torben Hansen[13], Helen Cox[6], Mary Reilly[14], Alex Rossor[14], Rebecca J Brown[3], Simeon I Taylor[15], Duncan McHale[8], Martin Armstrong[8], Elif A Oral[16], Vladimir Saudek[1,2], Stephen O'Rahilly[1,2], Eamonn R Maher[2,17]*†, Bjørn Richelsen[18]*†, David B Savage[1,2]*, Robert K Semple[1,2]*

[1]The University of Cambridge Metabolic Research Laboratories, Wellcome Trust-MRC Institute of Metabolic Science, Cambridge, United Kingdom; [2]The National Institute for Health Research Cambridge Biomedical Research Centre, Cambridge, United Kingdom; [3]National Institute of Diabetes and Digestive and Kidney Diseases, National Institutes of Health, Bethesda, United States; [4]Department of Public Health, Experimental and Forensic Medicine, University of Pavia, Pavia, Italy; [5]Institute of Cancer and Genomic Sciences, University of Birmingham, Birmingham, United Kingdom; [6]West Midlands Medical Genetics Department, Birmingham Women's Hospital, Edgbaston, Birmingham, United Kingdom; [7]Department of Endocrinology and Internal Medicine and Department of Clinical Epidemiology, Aarhus University Hospital, Aarhus, Denmark; [8]New Medicines, UCB Pharma, Slough, United Kingdom; [9]Wellcome Trust Sanger Institute, Cambridge, United Kingdom; [10]Wolfson Brain Imaging Centre, University of Cambridge School of Clinical Medicine, Cambridge Biomedical Campus, Cambridge, United Kingdom; [11]National Institute for Health Research/Wellcome Trust Clinical Research Facility, Cambridge University Hospitals NHS Foundation Trust, Cambridge Biomedical Campus, Cambridge, United Kingdom; [12]Cambridge University Hospitals NHS Foundation Trust, Cambridge Biomedical Campus, Cambridge, United Kingdom; [13]The Novo Nordisk Foundation Center for Basic Metabolic Research, Faculty of Health and Medical Sciences, University of Copenhagen, Copenhagen, Denmark; [14]MRC Centre for Neuromuscular Diseases, National Hospital for Neurology and Neurosurgery, UCL Institute of Neurology, London, United Kingdom; [15]University of Maryland School of Medicine, Baltimore, United States; [16]Metabolism, Endocrinology and Diabetes (MEND) Division, Department of Internal of Medicine, Brehm Center for Diabetes, Ann Arbor, United States; [17]Department of Medical Genetics, University of Cambridge School of Clinical Medicine, Cambridge, United Kingdom; [18]Department of Endocrinology and Internal Medicine, Aarhus University Hospital and Clinical Medicine, Aarhus University, Aarhus, Denmark

*For correspondence: erm1000@medschl.cam.ac.uk (ERM); bjoern.richelsen@aarhus.rm.dk (BR); dbs23@medschl.cam.ac.uk (DBS); rks16@cam.ac.uk (RKS)

†These authors contributed equally to this work

**Abstract** MFN2 encodes mitofusin 2, a membrane-bound mediator of mitochondrial membrane fusion and inter-organelle communication. MFN2 mutations cause axonal neuropathy, with associated lipodystrophy only occasionally noted, however homozygosity for the p.Arg707Trp mutation was recently associated with upper body adipose overgrowth. We describe similar massive adipose overgrowth with suppressed leptin expression in four further patients with biallelic MFN2 mutations and at least one p.Arg707Trp allele. Overgrown tissue was composed of normal-sized, UCP1-negative unilocular adipocytes, with mitochondrial network fragmentation, disorganised cristae, and increased autophagosomes. There was strong transcriptional evidence of mitochondrial stress signalling, increased protein synthesis, and suppression of signatures of cell death in affected tissue, whereas mitochondrial morphology and gene expression were normal in skin fibroblasts. These findings suggest that specific MFN2 mutations cause tissue-selective mitochondrial dysfunction with increased adipocyte proliferation and survival, confirm a novel form of excess adiposity with paradoxical suppression of leptin expression, and suggest potential targeted therapies.

## Introduction

Adipose tissue is critical for metabolic homeostasis, and either global excess (obesity) or pathologic deficiency (lipodystrophy) lead to metabolic disease. Anatomically distinct adipose depots vary in embryological origins, gene expression profiles and patterns of hormonal regulation (*Tchkonia et al., 2013*), but many questions about the determinants of adipose depot differences, and their importance for metabolic regulation, remain unanswered. In studies of human metabolic disease, attention focussed for many years on visceral white adipose tissue, and on femorogluteal subcutaneous white adipose tissue, however interest has recently increased sharply in upper body subcutaneous adipose tissue, especially in interscapular and supraclavicular regions, where both obligate and facultative thermogenic adipocytes have been shown to exist under some circumstances (*Betz and Enerbäck, 2015*).

Since the mid 19th century a devastating but rare disease characterised by a selective, massive and sometimes life-threatening increase in upper body adipose tissue has been known (*Brodie, 1846*; *Herbst, 2012*; *Madelung, 1888*). This disease is called either multiple symmetrical lipomatosis (MSL) or Madelung's disease. Although the degree of adipose overgrowth seen may be spectacular, it varies in distribution (*Herbst, 2012*) and the underlying pathogenesis, which is currently poorly understood, is likely to be heterogeneous. Previous studies have suggested hyperplasia of adipocytes which manifest impaired catecholamine-induced lipolysis (*Enzi et al., 1977*), and later structural studies of pathologically increased adipose tissue have suggested that it is composed of adipocytes with some morphological and metabolic features of dysfunctional brown adipocytes (*Enzi et al., 2015*). However MSL is not precisely defined, and phenotypic heterogeneity is apparent. People with MSL frequently exhibit variable additional features including peripheral neuropathy, autonomic neuropathy, non-alcoholic fatty liver disease and/or alcoholic liver disease, dyslipidaemia, and type 2 diabetes (*Herbst, 2012*). In some patients upper body adipose overgrowth contrasts with concomitant lipodystrophy in the femorogluteal region (*Herbst, 2012*). Physiological studies have described elevated resting energy expenditure, altered lipolytic rates and perturbed adipokine levels, however these findings have tended to vary, consistent with pathogenic heterogeneity (*Herbst, 2012*).

Sporadic MSL has been strongly linked to chronic alcohol excess (*Enzi et al., 2015*), and in rare cases is seen in the mitochondrial myoclonus epilepsy and ragged red fibers (MERRF) syndrome, accounted for in most cases by the mitochondrial tRNA(Lys) A8344G mutation (*Chong et al., 2003*). These observations led to the suggestion that mitochondrial dysfunction may be an important pathogenic factor in MSL. In 2015, a homozygous missense mutation, p.Arg707Trp, was identified in MFN2, or mitofusin 2, in three patients with MSL and neuropathy from two families (*Sawyer et al., 2015*). Mitofusin 2 is a nucleus-encoded mitochondrial outer membrane protein involved in mitochondrial fusion, mitochondrial interactions with the endoplasmic reticulum (*Naon et al., 2016*), and other aspects of cellular metabolism (*de Brito and Scorrano, 2008*; *Zorzano et al., 2015*). Arg707 lies in the carboxy-terminal coiled coil domain HR2 of mitofusin 2 that is believed to be critical for

**eLife digest** Obesity and the diseases associated with it are among the biggest healthcare problems in developed countries. The word obesity means, simply, the accumulation of too much fat tissue in the body, but this ignores growing evidence that fat tissue is highly complex. Fat tissue is important for "mopping up" and storing excess calories safely, but also sends messages to the brain and other organs to report how full the body's energy stores are. Understanding how fat tissues perform these roles will aid the development of strategies to treat or prevent obesity.

A hormone called leptin acts as a signal of the status of the body's fat stores. High levels of leptin in the blood tell the brain that the body has plenty of fat stored. On the other hand, if the levels of leptin in the blood become very low it tells the brain to prioritize finding food and shut down any nonessential processes. This helps to prevent the body from starving. It is not clear how the production of leptin is controlled, in part because fat tissues in different parts of the body behave very differently.

Individuals who have a particular rare genetic mutation accumulate large amounts of fat tissue in their upper bodies and gradually lose fat tissue in their arms and legs. Despite accumulating a lot of fat tissue in the upper body, these individuals have extremely low levels of leptin in their blood. To investigate this genetic condition, Rocha et al. studied two children with the mutation and their healthy parents.

The experiments show that this mutation alters a protein called mitofusin 2, which is found in cell compartments called mitochondria. Mitofusin 2 helps the mitochondria to bind to each other and to other parts of the cell, which is important for the mitochondria to generate the energy needed for vital cell processes. The mitochondria in the fat cells of the children are less closely linked to each other and have an unusual appearance compared to the mitochondria in the parents' fat cells. Further experiments showed that some genes, including the one that produces leptin, are less active in the children compared to their parents – while other genes that are involved in starvation or stress responses are more active.

This work suggests that mitochondria play an important role in regulating the production of leptin. Furthermore, it suggests that leptin or drugs that switch off stress-related genes may have the potential to be used to treat individuals with this particular mutation.

homotypic and heterotypic interactions of mitofusin 2 (*Franco et al., 2016*; *Koshiba et al., 2004*) with counterparts on other organelles, leading to the suggestion that the p.Arg707Trp mutation may compromise outer mitochondrial membrane fusion (*Sawyer et al., 2015*).

MFN2 p.Arg707Trp is the only reported cause of Mendelian MSL to date, however the nature of the associated adipose pathology, and the mechanism linking this mutation to MSL is unknown. We now describe four further patients with MSL associated with biallelic MFN2 mutations, in all cases including at least one p.Arg707Trp allele, investigate key aspects of their intermediary metabolism, and examine the microscopic and transcriptomic derangements in affected adipose tissue. We thereby provide a monogenic example of primary, apparently tissue-selective mitochondrial dysfunction leading to insulin resistance, and describe a form of leptin deficiency in the face of severe upper body obesity. Our findings yield novel insights into the role of mitofusin 2 in adipocyte energy metabolism and proliferation in humans *in vivo*, and suggest targeted treatments for MSL that warrant further assessment.

## Results

### Case histories

Four patients of European ancestry from three families were studied. The first patient (P1) presented to medical attention at 5 years old with increasing upper body adiposity. Despite repeated attempts to reduce food intake this progressed throughout childhood in tandem with loss of lower body adipose tissue. She also developed axonal peripheral neuropathy and secondary foot contractures, requiring corrective surgery at the age of 17 years. By the age of 18 years, her body mass index

(BMI) was 33 kg/m$^2$ due to massive upper body adiposity, which contrasted with near absence of femorogluteal, leg and distal arm adipose tissue (*Figure 1A*). She also had primary amenorrhea with clinical, biochemical and imaging-based evidence of partial hypogonadotropic hypogonadism including retarded bone age, delayed development of secondary sexual characteristics, and a small uterus, all in the context of relatively low luteinizing hormone (LH), follicle stimulating hormone (FSH) and estradiol levels (*Table 1*).

The parents of P1 (aged 43 and 51 years) were clinically unaffected. Exome sequencing in the family trio was undertaken and directly interrogated to look for mutations in MFN2, given the known association of MFN2 with axonal neuropathy, and its recently reported association with MSL (*Sawyer et al., 2015*). This revealed compound heterozygous mutations (p.Arg707Trp and p. Arg343del) in *MFN2* (MIM*608507) in P1, whereas her parents were single heterozygotes.

Patients P2 and P3 were more mildly affected. These patients, a 17-year-old female (P2) and a 16-year-old male (P3), were siblings from a multiply consanguineous family of Irish ancestry. Only the two affected siblings and their unaffected parents were available for genetic study, however multiple family members across three generations of the family were reported to have developed progressive overgrowth of adipose tissue in the upper back and chest, head and neck, from early in their second decades. Two older relatives required long term tracheostomies due to airway occlusion by overgrown neck adipose tissue, with MRI in one demonstrating diffuse fatty infiltration of the tongue and a lipomatous mass in the supraclavicular fossa. A paternal uncle was reported to have died perioperatively after airway problems related to adipose overgrowth in his fourth decade, while three further relatives were reported to show a similar pattern of head and neck adipose overgrowth. One of these had been demonstrated to have severe sensorimotor axonal neuropathy (*Figure 1—figure supplement 1*).

In keeping with the familial pattern of overgrowth, P2 and P3 developed striae, or stretch marks, over their upper back, together with adipose overgrowth and concomitant loss of fat from the limbs around the age of 13 years old. Examination of P2 at 16 years old showed mild loss of adipose tissue in the forearms, with abnormal adipose tissue accumulation around the neck, upper back, and upper arms. P3, examined at the age of 15 years old was more mildly affected, but showed dorsal striae, mild adipose tissue loss from the forearms, a prominent dorsal fat pad or 'buffalo hump' and increased adipose tissue in the anterior neck. Neither sibling showed any other clinical abnormalities, with age-appropriate progression through puberty, and normal nerve conduction studies.

Given parental consanguinity for all affected family members, an autosomal recessive mode of inheritance was hypothesized. Exome sequencing was undertaken for P2, P3 and their parents, and the p.Arg707Trp mutation in *MFN2* was identified in homozygous form in both P2 and P3, and in heterozygous form in both parents.

The final patient, P4, was evaluated at 17 and 37 years old. She first reported increased head and neck adiposity with increased muscularity of her extremities around 13 years old. This progressed over ensuing years, with striking accumulation of upper back and anterior neck adipose tissue associated with severely increased appetite and frequent food craving behaviours. Other symptoms included male pattern hair growth, an irregular menstrual cycle, and acanthosis nigricans, all suggestive of insulin resistance, as well as leg cramps after exercise, and burning and tingling sensations in hands and feet. On evaluation at 17 years old she was overweight with highly dysmorphic accumulation of truncal, head and neck adiposity (*Figure 1B*), extending to the pubic region. This contrasted with a paucity of adipose tissue on the limbs, which were muscular with prominent superficial veins. Adipose tissue removed surgically for cosmetic purposes was noted to be highly vascular and lobulated. She was subsequently lost to follow up until 37 years old. At that stage her dominant problems were distressing upper body adipose accumulation with back pain exacerbated by scoliosis, with leg cramps as well as burning and tingling in hands and feet. Menstrual bleeds were irregular, but she had had one successful pregnancy. Abnormally increased appetite persisted, but was less striking than during adolescence.

Sanger sequencing of *MFN2* was undertaken in P4 and her unaffected father, although her mother (not knowingly related to her father) was unavailable. P4 was found to be homozygous for the MFN2 p.Arg707Trp mutation, while her father was heterozygous.

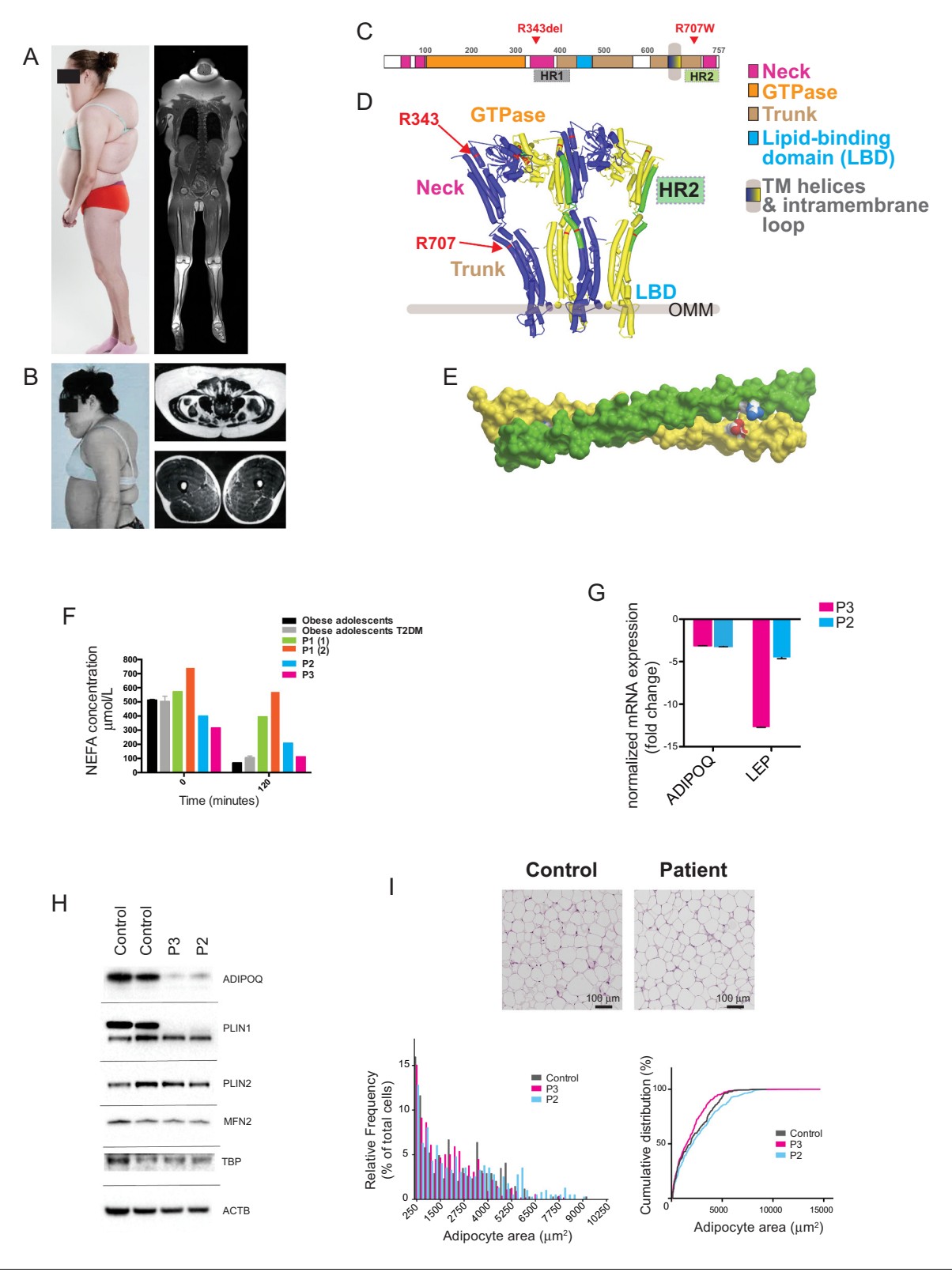

**Figure 1.** Multiple symmetrical lipomatosis associated with biallelic mutations in MFN2. (**A**) Profile of P1, showing upper body adipose overgrowth and loss of adipose tissue and muscle from legs (left) and T1-weighted coronal MRI image of P1, highlighting excess upper body fat (right). (**B**) Profile of P4, showing upper body adipose overgrowth (left) and T1-weighted MRI images highlighting excess lower abdominal adipose tissue and a paucity of adipose tissue at the mid femoral region. (**C**) Linear depiction of human MFN2, highlighting regions contributing to key structural domains. (**D**)

*Figure 1 continued on next page*

*Figure 1 continued*

Structure of a GTP-bound, extended tetramer (dimer of dimers) of Bacterial Dynamin-Like Protein (BDLP) as a model of human MFN2. Monomers are distinguished in yellow and blue and HR2 domain in green. The MFN2 mutations are shown in red. (**E**) Homology model of antiparallel homodimer of HR2 domains rendered in surface presentation with the individual monomers coloured in yellow and green and the charged atoms of residues R707 and D742 in blue and red. (**F**) Plasma non-esterified fatty acid (NEFA) concentrations in P1 (at age 18 (1) and 20 (2) years), P2 and P3 before and after a 75g oral glucose load. Data from 770 obese adolescents and 22 obese adolescents with type 2 diabetes (*Hershkop et al., 2016*) are shown as controls. (**G**) Quantitative real-time PCR of interscapular adipose biopsies from P2 and P3, expressed as fold change from five female controls. (**H**) Immunoblots from patient and control adipose lysates for adiponectin (ADIPOQ), perilipin 1 (PLIN1), perilipin 2 (PLIN2), and mitofusin 2 (MFN2). TATA-binding protein (TBP) and β-actin (ACTB) were used as loading controls. (**I**) Assessment of adipocyte size in interscapular adipose tissue. Representative haematoxylin and eosin-stained interscapular adipose tissues from a patient and a control are shown on top. Scale bars = 100 μm. The bottom left chart shows adipocyte size distribution analysis with bars indicating percentages of cells in the specified area range. The chart on the right shows cumulative frequencies. ≥500 adipocytes analyzed per biopsy.

The following figure supplements are available for figure 1:

**Figure supplement 1.** Pedigree diagram for P2, P3.

**Figure supplement 2.** GTP/GDP-dependent conformations of MFN2 modelled on homology to Bacterial Dynamin-Like Protein.

**Figure supplement 3.** Expression of perilipin in affected adipose tissue.

**Table 1.** Biochemical profiles of probands P1-4. MRS = Magnetic Resonance Spectroscopy; USS = Ultrasound scanning.

| | P1 | P2 | P3 | P4 | Reference range |
|---|---|---|---|---|---|
| Age at time of assessment - years | 18 | 17.8 | 16.7 | 17 | |
| Age at first presentation - years | c.5 | c.12 | c.13 | c.13 | |
| Height – m | 1.63 | 1.58 | 1.62 | 1.63 | |
| Weight – kg | 88.4 | 61.6 | 53.6 | 72 | |
| BMI - kg/m$^2$ | 33.0 | 24.7 | 20.4 | 27.0 | |
| Liver fat | 26% by MRS (normal < 5.5) | – | – | increased (USS and biopsy) | |
| Triglyceride - mmol/L | 7.1 | 1.0 | 0.9 | 7.7 | <1.7 |
| HDL-Cholesterol - mmol/L | 0.9 | 1.4 | 0.9 | 0.6 | >1.0 |
| Total-Cholesterol - mmol/L | 5.8 | 4.5 | 4.2 | 4.0 | <5.2 |
| Insulin - pmol/L | 89 | 96 | 152 | 264 | <60 |
| Glucose - mmol/L | 4.5 | 4.6 | 5.2 | 4.5 | <6.1 |
| Glycated haemoglobin -mmol/mol | 27 | 30 | 32 | 35 | <53 |
| Leptin - μg/L | 0.6 (14.9–60.2) | 2.4 (2.4–24.4) | 2.2 (0.4–8.3) | <0.6 (8.6–38.9) | * |
| Adiponectin - mg/L | 0.7 (2.6–14.9) | 0.8 (4.4–17.7) | 1.4 (2.6–12.6) | 2.0 (5.0–28.8) | * |
| Lactate – mmol/L | 7.3 | – | – | 4.8 | |
| LH – U/L | 2.0 | 10.1 | 5.3 | 3.0 | 1.3-8.4[‡] |
| FSH – U/L | 3.4 | 8.0 | 5.0 | 1.0 | 2.9-8.4[‡] |
| Estradiol – pmol/L | 105 | 107 | – | 35 ng/dl | 72-529[‡] |
| Testosterone – nmol/L | – | – | 7.6 | 93 ng/dl | 9-22[†] |

*Sex, BMI and assay-specific reference ranges are indicated in brackets for each patient.

[†]Adult male range.

[‡]Follicular phase.

## Mitofusin 2 mutations

Arg707 is exquisitely conserved among MFN2 orthologues, and lies in the second heptad repeat (HR2) coiled-coil domain (*Figure 1C,D*, *Figure 1—figure supplement 2*). The p.Arg707Trp variant is found with an allele frequency of 0.055% in Europeans in the ExAC dataset. Literature review revealed a further patient with MSL and the p.Arg707Trp mutation co-inherited with a second, different loss-of-function allele featuring deletion of exons 7 and 8 (*Carr et al., 2015*). Three further siblings were identified with p.Arg707Trp together with the p.Gly109Arg mutation. All three had severe early onset axonal neuropathy, but no adipose phenotype was reported when the oldest of the siblings was 25 years old (*Table 2*). No simple heterozygote for the p.Arg707Trp allele has been described with MSL, and most of those heterozygotes reported do not have clinically manifest neuropathy. No other HR2 domain mutations (at or distal to amino acid residue 646) were identified in either homozygous or compound heterozygous forms.

The structure of a full-length mitofusin is not yet known. Sequence homology searching using HHPred (*Söding, 2005*) identified mitofusins as the closest mammalian homologues of the bacterial dynamin-like protein (BDLP), whose structure has been solved (*Low et al., 2009*). Homology modeling suggests that mitofusins retain all key structural domains of BDLP (*Figure 1C*), and so mitofusin 2 may exist, like BDLP, in two conformations, one GDP-bound, lipid-free and compacted (*Figure 1—figure supplement 2A*) and one GTP-bound, extended and membrane-associated (*Figure 1D*). The extended conformation is expected to *cis*-dimerise upon GTP-binding, with dimers further *cis*-oligomerising on the cell membrane (*Low and Löwe, 2010*). In-frame deletion of Arg343 destroys the helix next to the hinge between the GTPase and stalk domains (*Figure 1D*) and is predicted to disrupt the tip of the stalk and seriously impair transition between closed and extended conformations.

Membrane fusion depends on trans-dimerization of mitofusins on different membranes. This requires a distinct GTPase-independent extended conformation featuring an anti-parallel dimer between their extended C-terminal domains HR2 (*Franco et al., 2016*). The X-ray structure of a

**Table 2.** Genotypes and phenotypes of all patients identified with biallelic MFN2 mutations including p.Arg707Trp; MSL = Multiple Symmetrical Lipomatosis.

| ID | Age, years | Sex | MFN2 genotype | Phenotype(s) (age of onset, years) | Comment | Reference |
|---|---|---|---|---|---|---|
| P1 | 18 | F | p.Arg707Trp R343del | MSL (<10) Axonal neuropathy (<10) | ↑↑ lactate ↓↓ leptin ↓↓ adiponectin | This study |
| P2 | 17 | F | p.Arg707Trp p.Arg707Trp | MSL (c.13) No neuropathy (17) | ↓ adiponectin | This study |
| P3 | 16 | M | p.Arg707Trp p.Arg707Trp | MSL (c.13) No neuropathy (16) | ↓ adiponectin | This study |
| P4 | 37 | F | p.Arg707Trp p.Arg707Trp | MSL (c.13) Likely neuropathy* Renal tubular dysfunction[†] (c.13) | ↑↑ lactate ↓↓ leptin ↓↓ adiponectin | This study |
| Sawyer *et al* Patient 1 | 61 | M | p.Arg707Trp p.Arg707Trp | MSL (20 s) Axonal neuropathy (50 s) Diabetes (50 s) | ↑ lactate ↓↓ leptin ↓ adiponectin | (*Sawyer et al., 2015*) |
| Sawyer *et al* Patient 2 | 63 | M | p.Arg707Trp p.Arg707Trp | MSL (40 s) Axonal neuropathy (40s) | ↑ lactate | (*Nicholson et al., 2008*; *Sawyer et al., 2015*) |
| Sawyer *et al* Patient 3 | 60 | F | p.Arg707Trp p.Arg707Trp | MSL (20 s) Axonal neuropathy (c.2) | ↑ lactate | (*Sawyer et al., 2015*) |
| Carr *et al* Patient 4/II-1 | 44 | M | p.Arg707Trp MFN2 ex7-8del | MSL (not reported) Axonal neuropathy (2) | | (*Carr et al., 2015*) |
| Calvo *et al* Patient 6 | 19 | M | p.Arg707Trp p.Gly108Arg | Axonal neuropathy (<10) | | (*Calvo et al., 2009*) |
| Calvo *et al* Patient 7 | 24 | F | p.Arg707Trp p.Gly108Arg | Axonal neuropathy (<10) | | (*Calvo et al., 2009*) |
| Calvo *et al* Patient 8 | 25 | M | p.Arg707Trp p.Gly108Arg | Axonal neuropathy (<10) | | (*Calvo et al., 2009*) |

*Based on clinical signs and symptoms.

[†]increased renal protein, magnesium, potassium and calcium with low serum potassium and magnesium; still apparent at 37 years old.

MFN1 HR2 fragment in a homodimer is known (*Koshiba et al., 2004*) and the structure of the corresponding MFN2 homodimer (*Figure 1E*) and MFN1/MFN2 heterodimer (not shown) can readily be modelled. The HR2 domain is involved in different interfaces in GDP-bound monomers (*Figure 1—figure supplement 2B*), in GTP-bound *cis*-oligomers (in which there are two different interfaces between dimers and between dimer of dimers [*Figure 1D*]), in GTPase-independent *trans*-dimers (*Figure 1E*) and in MFN2/MFN1 *trans* heterodimers. Arg707 contributes differently to the stability of these conformations: whereas it makes an ion pair with Asp742 in the trans antiparallel homodimer (*Figure 1E*) or with Glu728 of MFN1 in the heterodimer, it appears on the surface in all other conformations and is not directly involved in intra- or intermolecular helix packing. Its mutation will thus almost certainly destabilise trans-dimerisation by destroying the ion pair, but will influence other conformations to much lesser degrees.

## Adipose tissue dysfunction

Given the anatomically obvious increase in upper body adipose tissue in all four patients studied, the function of that adipose tissue was next assessed *in vivo* and *ex vivo*. Fasting blood tests showed P1 and P4 but not P2 and P3 to have elevated plasma triglycerides, while P1 and P4 also had severe hepatic steatosis (*Table 1*). Initial oral glucose tolerance testing revealed normal glucose tolerance with elevated insulin levels in all four patients, consistent with moderate to severe insulin resistance, with fasting hyperglycaemia seen in P1 and P4 at 20 and 37 years respectively (*Table 3*). P1 and P4 also showed failure of the normal suppression of plasma non-esterified fatty acids (NEFA) in the face of increased plasma insulin. This was assessed by OGTT at two ages in P1, in whom impaired NEFA suppression contrasted both with healthy controls and with obese adolescents with overt type 2 diabetes (*Hershkop et al., 2016*) (*Figure 1F*). In P4 it was assessed by frequently sampled intravenous glucose tolerance testing using a previously published protocol (*Sumner et al., 2004*), and basal NEFA of 786 µmol/L showed maximal suppression of only 18%, occurring 27 min after the glucose challenge. These findings suggest impaired suppression of lipolysis by insulin in the overgrown adipose tissue and/or the depleted lower limb adipose depots of P1 and P4. Blood lactate concentration was elevated at baseline and on two occasions in P1, rising further after an oral glucose load, suggesting impairment of mitochondrial function in an insulin-responsive tissue, although whether adipose tissue, muscle or both were responsible for this could not be discerned (*Table 3*). Lactate levels were not determined in P2 and P3.

Serum adipokine profiles were highly abnormal in all four patients. Serum leptin was undetectable in P1 and P4 on initial evaluation (*Table 1*) despite increased upper body adiposity, as evidenced by

**Table 3.** Oral glucose tolerance test results for patients P1, P2 and P3. P1 and P4 were tested on two occasions.

|  | Time (minutes) | 0 | 30 | 60 | 90 | 120 |
|---|---|---|---|---|---|---|
| P1 (18 years old) | Glucose (mmol/L) | 4.6 | 6.4 | 6.4 | 4.3 | 4.6 |
|  | Insulin (pmol/L) | 109 | 1010 | 996 | 249 | 190 |
| P1 (20 years old) | Glucose (mmol/L) | 6.9 | 9.3 | 9.8 | 9.7 | 9.3 |
|  | Insulin (pmol/L) | 207 | 630 | 609 | 677 | 708 |
|  | Lactate (mmol/L) | 6.8 | 8.3 | 9.6 | 9.9 | 9.9 |
| P2 (17.8 years old) | Glucose (mmol/L) | 4.6 | 7.1 | 8.9 | 6.0 | 6.2 |
|  | Insulin (pmol/L) | 96 | 595 | 962 | 478 | 566 |
| P3 (16.7 years old) | Glucose (mmol/L) | 5.7 | 9.8 | 8.4 | 7.0 | 5.5 |
|  | Insulin (pmol/L) | 199 | 2048 | 1796 | 1888 | 773 |
| P4 (17 years old) | Glucose (mmol/L) | 4.5 | 5.8 | 6.9 | 5.8 | 6.3 |
|  | Insulin (pmol/L) | 264 | 1160 | 2479 | 1361 | 1514 |
| P4 (37 years old) | Glucose (mmol/L) | 6.6 | – | – | – | – |
|  | Insulin (pmol/L) | 688 | – | – | – | – |

body mass indices of 33 and 27 kg/m$^2$ and whole body adipose content determined by DXA of 31% and 27.4% respectively. Serum leptin remained undetectable in P4 at 37 years old. In P2 and P3 circulating leptin concentrations were also low, but overlapped with levels seen in healthy controls, possibly reflecting the relatively early stage of evolution of the pathological adipose overgrowth. In all patients serum adiponectin levels were suppressed to levels usually only seen in generalised lipodystrophy or extreme, non receptoropathy insulin resistance (*Table 1*) (*Semple et al., 2006*). Leptin secretion from adipose explants taken from the neck and abdominal subcutaneous adipose tissue of P1 was moreover undetectable, and leptin mRNA expression in the same sample was commensurately suppressed (data not shown). Surgical biopsies of overgrown interscapular subcutaneous adipose tissue were independently obtained from P2 and P3, and from five age-matched otherwise healthy female controls undergoing elective surgery for idiopathic scoliosis. Leptin mRNA expression, and adiponectin mRNA and protein expression were severely suppressed in P2 and P3 compared to controls, while mitofusin 2 protein levels were unchanged (*Figure 1G,H*). In view of the reduced suppression of lipolysis seen in P1, perilipin 1 and 2 expression were determined, with levels of perilipin 1 found to be reduced, whereas perilipin 2 expression was preserved (*Figure 1H*).

In all three patients from whom subcutaneous adipose tissue biopsies were obtained, histological examination revealed the expanded adipose tissue to feature exclusively unilocular adipocytes (*Figure 1I*). UCP1 immunostaining was negative (data not shown). There was no difference in adipocyte size among patients and controls (*Figure 1I*), and in keeping with low perilipin expression in affected adipose tissue detected by immunoblotting, immunostaining for perilipin was also reduced, with marked heterogeneity seen in patient but not control adipose tissue (*Figure 1—figure supplement 3*).

In contrast to the nearly normal appearance of affected adipose tissue by light microscopy, electron microscopy of adipose tissue from P2 and P3 revealed highly abnormal adipocyte and preadipocyte ultrastructure. Control adipocytes exhibited scattered ovoid mitochondria with narrow cristae (*Figure 2A*), while adipocytes from P2 and P3 showed thickening of the cytoplasmic rim and proliferation of round, enlarged mitochondria, with fragmented cristae on cross sectional imaging (*Figure 2B*), reminiscent of those reported in different cell types and tissues in the context of Mfn2 deficiency (e.g. [*Lee et al., 2012*]). Membrane-bound structures enclosed in double membranes were seen in many cells (*Figure 2B,C,D*), potentially consistent with mitophagy, although no examples of morphologically distinct mitochondria enclosed in such membranes were seen. In occasional cells apparent extrusion of membrane-enclosed damaged mitochondria was seen (*Figure 2D*). In both P2 and P3, 3–4 preadipocytes were seen among approximately 30 adipocytes studied by electron microscopy, while no preadipocytes were seen in control samples. Preadipocytes were associated with capillaries, showed characteristic small lipid droplets and glycogen granules, but also showed evidence of proliferation of fragmented mitochondria, many with deranged cristae, as seen in mature adipocytes (*Figure 2D*).

mRNA extracted from affected adipose tissue from three patients was subjected to transcriptomic analysis, using RNAseq in the case of P2, P3 and five matched controls, and a microarray-based approach in a different laboratory in the case of P1, from whom subcutaneous adipose samples were independently studied from both neck and abdomen and compared to abdominal subcutaneous adipose tissue from ten female controls. Primary analysis was undertaken of P2 and P3 compared to controls, with the more limited data from the comparison of P1 with controls subsequently aligned. Affected adipose tissue was transcriptionally distinct from control adipose tissue (*Figure 3A*), and unsupervised pathway analysis identified the most highly significant perturbation of gene expression to be related to mitochondrial dysfunction and oxidative phosphorylation (*Figure 3B*). The mitochondrial signal was accounted for by downregulation of 66 of 68 transcripts derived from mitochondria-encoded genes, contrasting with upregulation of 83 of 91 nucleus-encoded mitochondrial genes (*Figure 3C*). Despite this upregulation, however, immunoblotting for a panel of nucleus-encoded oxidative phosphorylation proteins showed universally strongly reduced levels, with citrate synthase, a marker of mitochondrial mass, also sharply reduced (*Figure 3D*). Adipose mitochondrial DNA was modestly reduced in P2 and P3, but was not assayed in P1 (*Figure 3E*). These findings suggest that the increased numbers of morphologically abnormal mitochondria seen on electron microscopy in P2 and P3 are severely hypofunctional, and that the increased transcript levels of nucleus-encoded mitochondrial genes represents a compensatory response that is insufficient to normalise function. Indeed, transcript levels of *PGC1A*, *PGC1B*, and

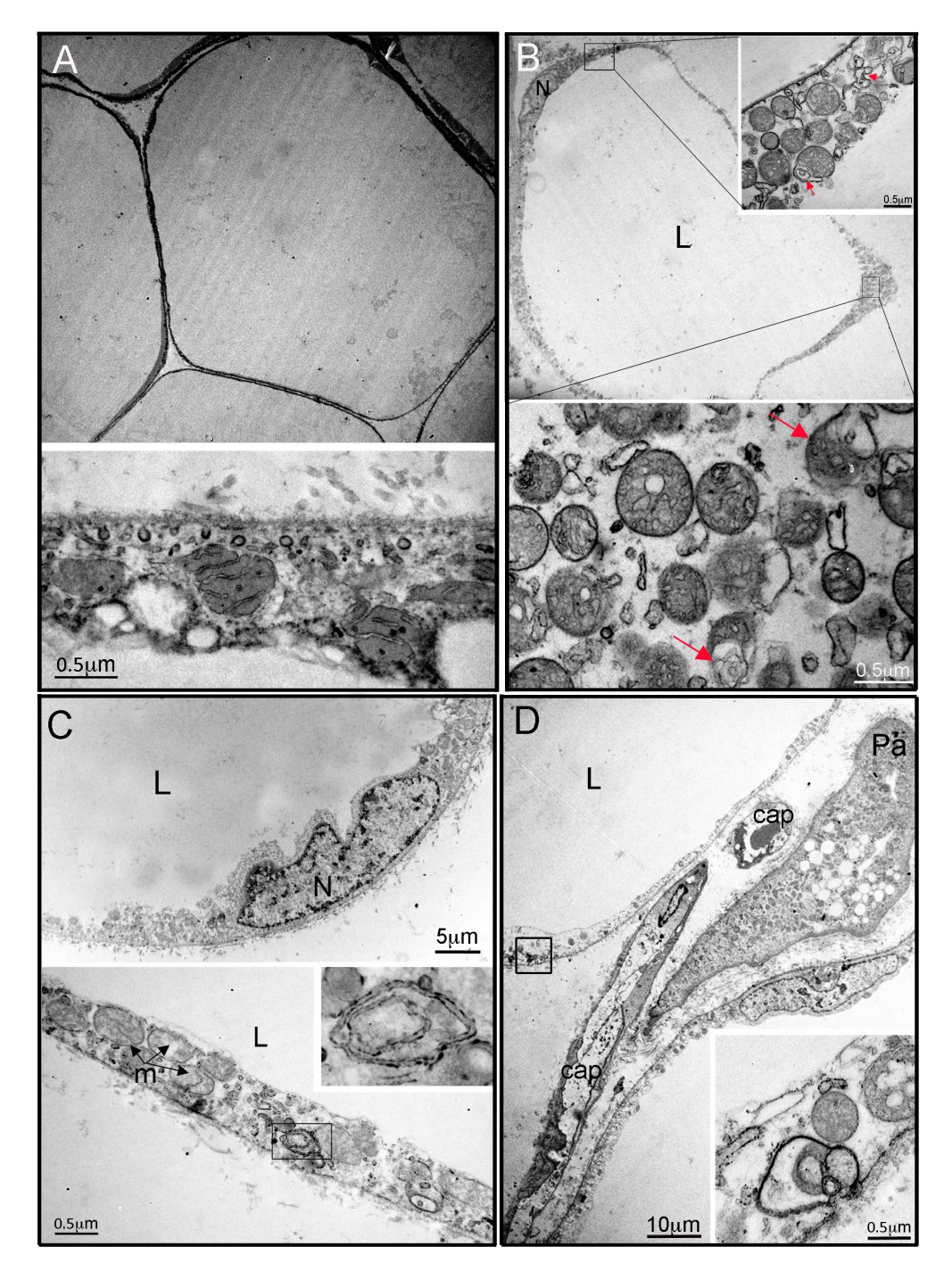

**Figure 2.** Ultrastructural appearances of interscapular subcutaneous adipose tissue in individuals with the homozygous MFN2 p.Arg707Trp mutation. (**A**) Transmission electron micrograph (TEM) of interscapular subcutaneous adipose tissue obtained from a control volunteer, showing a thin cytoplasmic rim, and, in the panel below, ovoid mitochondria with thin cristae. (**B**) TEM of interscapular subcutaneous adipose tissue obtained from P2, showing a thickened cytoplasmic rim containing increased numbers of round, enlarged mitochondria. Lower panel and inset: Mitochondria showed evidence of degeneration with disorganised cristae and complex inclusions (pointed by red arrows); L= lipid droplet. (**C**) TEM of affected adipose tissue showing mitochondria at different degree of degeneration (m) and double membrane-bound structures suggestive of mitophagy (inset in lower panel shows a concentric double membrane). N: adipocyte nucleus. (**D**) TEM of juxtacapillary cell representative of several pre-adipocytes (Pa) identified in P2 and P3 but in no controls. Early lipid accumulation and glycogen granules can be seen. In the inset, another image suggestive for active mitophagy is seen in the adjacent adipocyte.

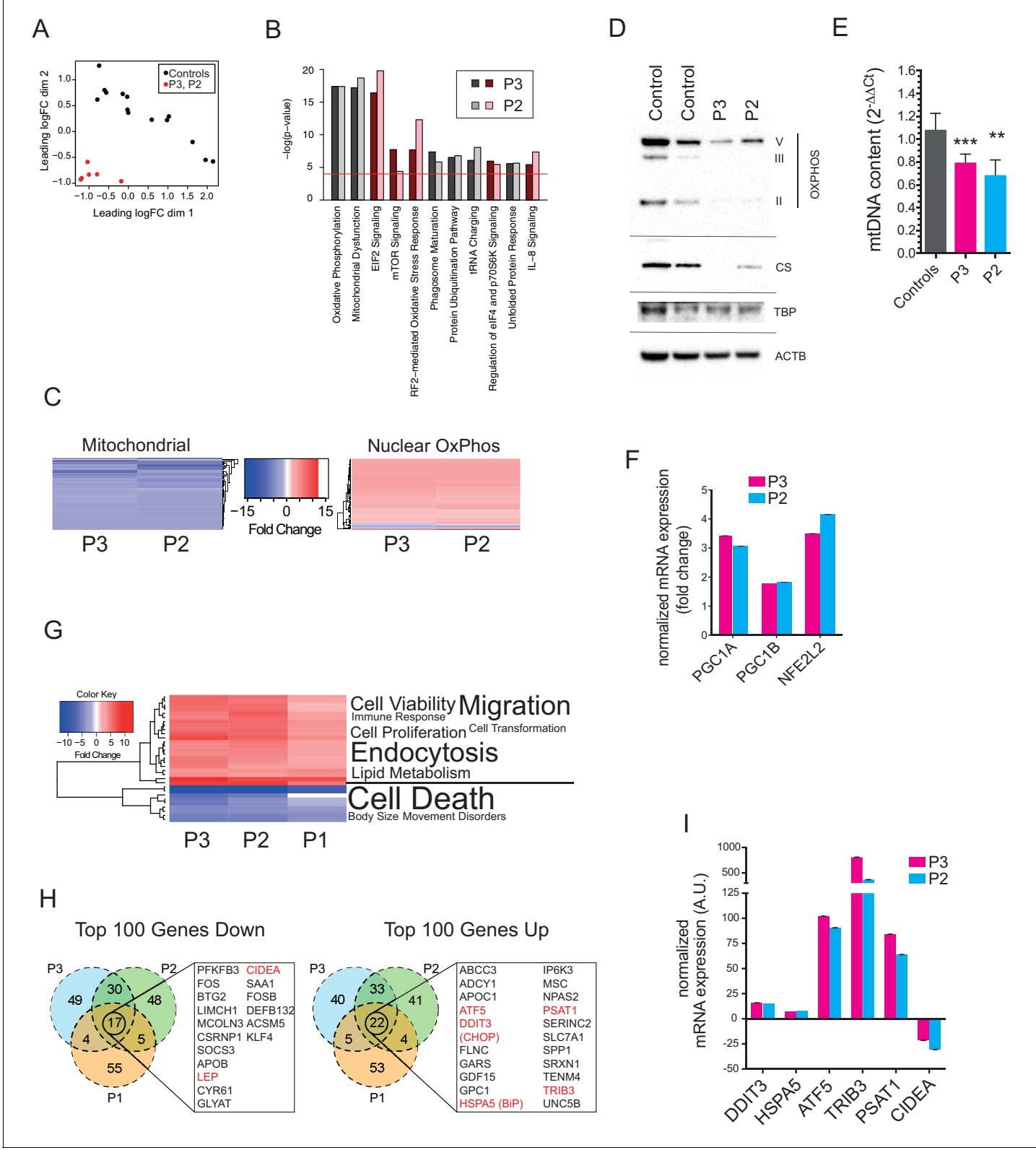

**Figure 3.** Gene expression profiling of hyperplastic adipose tissue. (**A**) Multidimensional Scaling Plot showing distinct patterns of gene expression of interscapular tissue samples from P2 and P3, compared to five age-matched controls. Three separate tissue samples from each of the patients and two to three separate tissue samples from the controls were used in the analysis. (**B**) Bar chart showing the 10 most significantly altered pathways identified by Ingenuity Pathway Analysis (IPA). Canonical pathways are shown in red to indicate increased expression; non-canonical pathways, to which no

*Figure 3 continued on next page*

*Figure 3 continued*

directionality was assigned, are shown in grey. The red line represents the p-value cut-off of <0.0001. (C) Heat maps showing decreased expression of genes in the mitochondrial genome and increased expression of nuclear-encoded oxidative phosphorylation (OxPhos) genes in P2 and P3. (D) Immunoblots of control and patient adipose tissue for OXPHOS subunits ATP synthase subunit alpha (V), Complex III subunit Core 2 (III), Complex II subunit 30 kDa (II), and the mitochondrial marker citrate synthase (CS). TATA-binding protein (TBP) and β-actin (ACTB) levels were used as loading controls (as also shown in *Figure 1H*). (E) Reduced levels of mtDNA in affected adipose tissue determined by Taqman real-time PCR. Means of three independent experiments ± SEM are shown. (**p<0.01). (F) Increased expression of *PGC1A, PGC1B,* and *NEF2L2* assessed by real-time qPCR and expressed as fold change in expression relative to five controls. Shown are means of 3 replicates. Error bars represent SEM. (G) Heat map showing the top 32 'disease and biology' pathways from IPA. A word/tag weighted cloud of component pathways is shown. (H) Venn Diagrams comparing the top 100 genes up and down with an FDR cut-off of <0.00001 among patients. Genes shared between all three patients' samples are shown in expanded lists with genes noted in the text highlighted in red. (I) Transcript levels of *DDIT3, HSPA5, ATF5, TRIB3, PSAT1,* and *CIDEA* in P3 and P2, expressed as fold change relative to controls determined by Taqman quantitative real-time PCR. Shown are means of eight replicates of different dilutions with error bars calculated from R-squared values of fitted line.

The following source data and figure supplement are available for figure 3:

**Source data 1.** Differential expression analysis data.
**Source data 2.** Ingenuity Pathway Analysis results for P1.
**Source data 3.** Ingenuity Pathway Analysis results for P2.
**Source data 4.** Ingenuity Pathway Analysis results for P3.
**Source data 5.** Data used to generate the heat maps shown in *Figure 3C*.
**Source data 6.** Ingenuity Pathway Analysis 'Disease and Biology' data used to generate the heat map shown in *Figure 3G*.
**Figure supplement 1.** Expression of human brown, 'beige', and white adipocyte markers in affected adipose tissue.

*NFE2L2*, encoding transcription factors driving mitochondrial biogenesis were upregulated 3.2-fold, 1.8-fold, and 3.8-fold, respectively (*Figure 3F*).

Other genesets related to cellular stress responses were also highly significantly altered, including oxidative stress and unfolded protein response pathways, with additional evidence of mTOR pathway activation (*Figure 3B*). Despite the strong transcriptional suggestion of mitochondrial dysfunction and secondary activation of stress signalling pathways, analysis of disease-associated transcriptional pathways showed that signatures of cell and tissue death were suppressed, while signatures related to cell and tissue proliferation and survival were strongly upregulated (*Figure 3G*).

The top 100 upregulated and downregulated genes were then compared across all three patients studied. This revealed 22 shared significantly upregulated genes, which featured genes involved in the unfolded protein response, including the key transcription factors DDIT3 (better known as CHOP) and ATF5, the ER chaperone HSPA5 (better known as BiP), and TRB3, which has been shown in different contexts to inhibit insulin signalling (*Du et al., 2003*) and to drive apoptosis (*Ohoka et al., 2005*) among other roles (*Figure 3H,I*). BiP, ATF5, and TRB3 are known to be direct transcriptional targets of CHOP and ATF4 (*Han et al., 2013*), which was also highly significantly upregulated, while ATF5 has also recently been suggested to serve as a direct and critical mediator of the mammalian mitochondrial unfolded protein response (*Fiorese et al., 2016*), analogous to ATFS-1 in *C. elegans*. Strong transcriptional downregulation of both leptin and adiponectin was confirmed in all three patients, with other downregulated genes including the brown fat lipid droplet-associated gene CIDEA, and the pro-adipogenic transcription factor KLF4 (*Figure 3H,I*).

To address the type of adipose tissue in affected depots, expression of genes suggested in recent human studies to be markers for white, beige and brown fat were examined (*Basse et al., 2015*; *Carey et al., 2014*; *Lee et al., 2016*). No consistent pattern of change in gene expression was seen, with BAT- and beige adipose tissue-selective genes unchanged or lower in affected tissue than in controls, with the exception of CITED1, proposed as a marker of beige fat, which was upregulated in all three patients. WAT genes showed no consistent changes in expression from controls

(*Figure 3—figure supplement 1*), consistent with the overgrown adipose tissue studied being white adipose tissue.

## Dermal fibroblast studies

Dermal fibroblasts were cultured from the site of adipose biopsy in P1, P2, and P3. Like affected adipose tissue, cultured dermal fibroblasts showed normal MFN2 expression, however quite unlike in adipose tissue, expression of citrate synthase (*Figure 4A*) and a panel of oxidative phosphorylation proteins was normal (*Figure 4B*), while mitochondrial DNA content was mildly lower than in controls (*Figure 4C*). Furthermore, immunocytochemistry showed appropriate localisation of MFN2 expression to the mitochondrial network (*Figure 4D*, *Figure 4—figure supplement 1*), which in patient cells appeared morphologically indistinguishable from control cells both at baseline (*Figure 4E*), and after treatment with the small molecule mitochondrial fission inhibitor mdivi-1 (*Figure 4—figure supplement 2*). Transmission electron microscopy of dermal fibroblasts from P1-3 revealed normal mitochondrial ultrastructure, with no evidence of the striking fragmentation and dysmorphism seen in adipose biopsies (*Figure 4—figure supplement 3*). Diffusion of photoactivated GFP within the network was also normal when assessed in living cells (*Figure 4F*), although this diffusion was impaired in wild type cells by hydrogen peroxide treatment to fragment the network (data not shown). Mitochondrial membrane potential was unchanged in patient cells, as assessed by a mitochondrial membrane potential sensor JC-1 ratio assay (*Figure 4G*).

Neither adipose tissue nor dermal fibroblasts were available for study from P4, however electron microscopy of a liver biopsy obtained at 17 years old was reported to show increased lipid droplets and pericellular collagen but neither abnormal mitochondrial size nor ultrastructure. Together with dermal fibroblast findings this is consistent with the abnormalities seen in adipose tissue in this study being tissue-selective.

## Discussion

MSL is a phenotypically dramatic condition that has been known since the nineteenth century. Understanding of its pathogenesis is likely to yield insights into poorly understood aspects of adipose tissue function that are critical to pandemic diseases, including the determinants of the differing metabolic behaviours of different adipose depots. Study of MSL has been hampered, however, by its clinical heterogeneity, and lack of knowledge of its cause (*Herbst, 2012*). The recent report of homozygosity for the MFN2 p.Arg707Trp mutation in three individuals with MSL (*Sawyer et al., 2015*) suggested the first germline Mendelian cause of the disorder, presenting a critical opportunity to further understanding of the underlying pathogenesis, however the pathology of the overgrown adipose tissue and the systemic metabolic consequences were not specifically studied. We now describe four more affected patients, confirming the genetic association, demonstrate that increased adipose mass is accounted for by hyperplasia of UCP1-negative, unilocular adipocytes, and provide evidence of mitochondrial dysfunction in affected adipose tissue. We also show that depot-selective adipose hyperplasia is discordant with suppressed leptin levels and impaired insulin-mediated suppression of lipolysis in severely affected patients, while insulin resistance is seen in all. These findings offer a new window into the extraordinary depot-selective adipose overgrowth of MSL, and offer new opportunities for novel therapy, and to gain insights into fundamental aspects of adipocyte behaviour crucial to pandemic disease.

Bioenergetic efficiency and ATP-generating capacity of mitochondria are heavily influenced by the dynamic architecture of the mitochondrial network (eg. [*Liesa and Shirihai, 2013*; *Mishra and Chan, 2016*]). This is regulated by specialised proteins including mitofusin 2 (*Schrepfer and Scorrano, 2016*), which is important not only in mitochondrial fusion, but also in mitochondrial-ER tethering (*Naon et al., 2016*), apoptosis (*de Brito and Scorrano, 2008*), mitophagy (*Chen and Dorn, 2013*) and regulation of the unfolded protein response (*Pablo Muñoz et al., 2014*). Global knockout of Mfn2 in mice is embryonically lethal (*Chen et al., 2003*), but knockout specific to cardiac or skeletal muscle (*Chen and Dorn, 2013*; *Sebastián et al., 2016*), to dopaminergic neurones (*Pham et al., 2012*), to the renal proximal tubule (*Gall et al., 2015*), or to POMC neurones of the hypothalamus (*Schneeberger et al., 2013*) is tolerated. Nevertheless in each case cell or organ dysfunction is seen. Together with evidence that *MFN2* expression is lowered in key metabolic tissues in disease states (*Schneeberger et al., 2013*; *Zorzano, 2009*), this has led to keen interest in mitofusin 2 as a

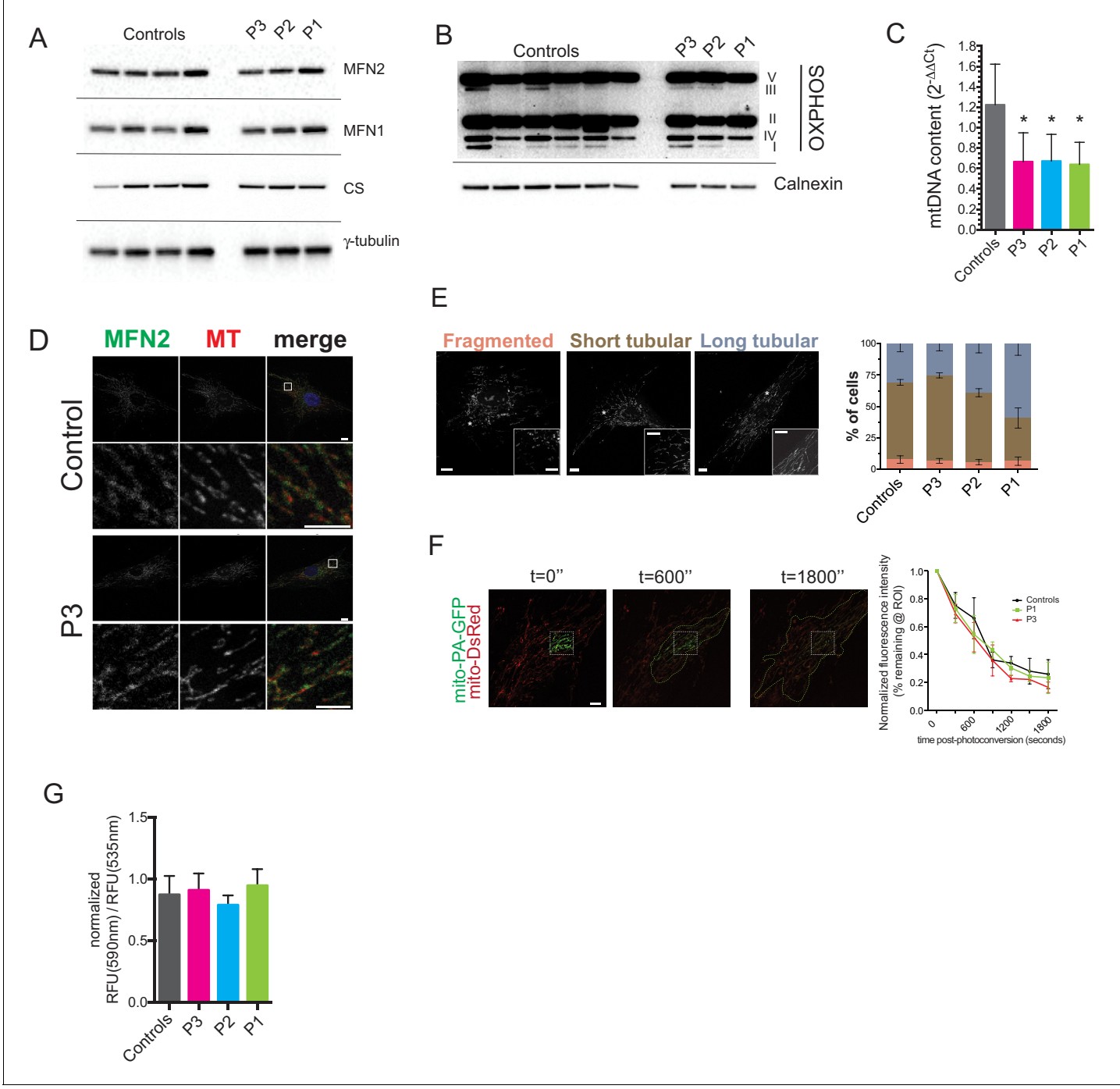

**Figure 4.** Preserved mitochondrial network in dermal fibroblasts from patients with MFN2-related MSL. (**A**) Immunoblots of dermal fibroblast lysates for mitofusin 2 (MFN2), mitofusin 1 (MFN1), and the mitochondrial marker citrate synthase (CS). γ-tubulin was used as loading control. (**B**) Immublotting of OXPHOS subunits using an optimized pre-mixed cocktail of monoclonal antibodies specific to Complex I subunit NDUF88 (CI), Complex II subunit 30 kDa (CII), Complex III subunit Core 2 (CIII), Complex IV subunit (IV), and ATP synthase subunit alpha (CV). (**C**) Reduced levels of mtDNA in patient dermal fibroblasts determined by Taqman real-time PCR. Means of three independent experiments ± SEM are shown. (*p<0.05). (**D**) Immunofluorescence confocal microscopy of endogenous MFN2 in fixed control and patient dermal fibroblasts showing preserved MFN2 expression and localization to mitochondria (See also *Figure 4—figure supplement 1*). Mitochondria were visualized with MitoTracker Orange (MT). Images in the bottom rows show digitally magnified detail of the boxed areas in the top rows. Scale bars: 10 μm (top rows); 5 μm (bottom rows). (**E**) Normal mitochondrial morphology in patient dermal fibroblasts demonstrated using the mitochondrial dye MitoTracker Orange. Cells were classified as having mostly fragmented, short tubular, and long tubular mitochondria as indicated in the representative confocal microscopy images shown. Asterisks indicate regions digitally magnified to show mitochondrial morphology in detail (insets). Scale bars, 10 μm. Data from four independent experiments

*Figure 4 continued on next page*

*Figure 4 continued*

(>50 cells per experiment) are presented as means ± SD. (F) Visualization and quantification of mitochondrial fusion using photoactivatable mitochondria matrix-targeted GFP (mito-PA-GFP): a dermal fibroblast from P1 expressing both mito-PA-GFP and mito-DsRed is shown to illustrate the *in vivo* mitochondria fusion assay. Photoconversion of PA-GFP was restricted to the area marked by the white dashed box. The activated mito-PA-GFP molecules, in time, redistributed along the area within the green dashed lines. The percentage of fluorescence remaining in the boxed region was measured, normalized to mito-DsRed signal, and plotted against time. Data are represented as means ± SD of single cell time-lapse measurements (10 randomly selected cells expressing both mito-DsRed and mito-PA-GFP per cell line were imaged). Scale bar, 10 µm. (G) Quantification of mitochondrial membrane potential (△Ψ) using the fluorescent sensor JC-1. Data show ratio of 590 nm (red) and 535 nm (green) fluorescence signals. Data from three independent experiments are represented as mean ± SD.

The following figure supplements are available for figure 4:

**Figure supplement 1.** Mitochondrial network morphology and mitofusin 2 expression in dermal fibroblasts from P1 and P2.

**Figure supplement 2.** Response of dermal fibroblasts to mitochondrial fission inhibitor mdivi-1.

**Figure supplement 3.** Transmission electron micrographs of dermal fibroblasts obtained from A. a control volunteer, B. P3, and C. P1, showing normal mitochondrial morphology in all cases.

---

contributor to the pathogenesis of pandemic obesity-related diseases. No adipose-specific knockout has been described.

Despite this largely murine evidence for important roles of MFN2 in several tissues, MFN2 loss-of-function mutations in humans have until recently been implicated solely in axonal sensorimotor neuropathy (*Züchner et al., 2004*), which is only weakly recapitulated in mouse models (*Detmer et al., 2008*; *Strickland et al., 2014*). Our findings, together with those of *Sawyer et al. (2015)*, now highlight a second tissue-selective phenotype in humans, namely severe upper body adipose overgrowth or 'Multiple Symmetrical Lipomatosis' (MSL). All cases of MFN2-related MSL have featured biallelic mutations including at least one MFN2 p.Arg707Trp allele, and many, but not all, patients have shown early-onset neuropathy in addition. Our structural analysis suggests that the mutation may have differential effects on different conformations of MFN2, particularly trans-dimer-isation and thus membrane fusion. It may be important that MFN2 homodimers will be most affected (*Figure 1E*) as both surfaces at the interface have a mutation, while in heterodimers only one surface is mutated. The second mutation we report in P1, who is severely affected, is predicted significantly to impair or to abrogate MFN2 function by destroying the helical support of the hinge between the GTPase and stalk domains (*Figure 1C*). Although the association of the MFN2 p. Arg707Trp mutation specifically with MSL is striking, existing human genetic data do not exclude the possibility that other HR2 domain mutations may also lead to the phenotype when both MFN2 alleles are affected.

The overgrown adipose tissue is unable to execute its key metabolic functions normally, as evidenced by insulin resistance, and reduced suppression of circulating free fatty acid concentrations after a glucose load. One caveat to this assertion is that it is possible that the 'lipodystrophic' or triglyceride-depleted lower limb adipose depots might also be contributing to the perturbed suppression of lipolysis and insulin resistance. Most strikingly, despite dramatic upper body adipose overgrowth in MFN2–related MSL, adipose tissue expression of the archetypal adipocyte hormones leptin and adiponectin was extremely low, with P1, P4 and one of the older, more severely affected patients previously described having undetectable plasma leptin despite high fat mass (*Sawyer et al., 2015*). We hypothesize that the differences observed in leptin concentrations among the patients we describe reflects the evolution of the disease, with all adipose depots affected either by overgrowth (upper body) or dystrophy (lower body) in P1 and P4, while in P2 and P3 some normal adipocytes remain. We suggest that advanced MFN2-related MSL represents a fourth form of severe leptin deficiency in humans, adding to congenital LEP mutations, lack of adipocytes in generalised LD, and 'empty' adipocytes in extreme leanness or starvation. We note that this phenotype may not be present in all MSL cases, as others have reported normal leptin levels in MSL cases of undefined cause (*Haap et al., 2004*).

Suppressed leptin expression could be explained if the pathologically expanding cell lineage had naturally low levels of leptin expression, as reported for brown adipocytes, or alternatively may reflect perturbation of a cellular mechanism sensing mitochondrial function and cellular energy state and transducing it into leptin gene expression. Although we demonstrate that increased adipose mass is accounted for by hyperplasia of cells with morphologic and transcriptional characteristics of UCP1-negative, unilocular white adipocytes, we cannot rule out the possibility that these cells are dysfunctional, 'whitened' thermogenic adipocytes, and indeed mitochondrial depletion has previously been shown to be involved in 'whitening' of beige adipocytes (*Altshuler-Keylin et al., 2016*).

Evidence of impaired mitochondrial function in MFN2-related MSL is seen in strikingly elevated blood lactate concentrations in P1, P4 and previously described patients (*Sawyer et al., 2015*), reduced expression of citrate synthase and respiratory chain components, and highly abnormal mitochondrial ultrastructure in affected adipose depots. The reduction in mitochondria-encoded transcripts, contrasting with increased nucleus-encoded mitochondrial transcripts and mitochondrial biogenesis factors suggests that compensatory transcriptional programmes are activated to try to restore cellular mitochondrial function. In contrast, gene expression, and mitochondrial morphology and fusion of primary dermal fibroblasts is normal, confirming mitochondrial dysfunction in MFN2-related MSL to be tissue-selective, and suggesting that MFN2 Arg707Trp remains able to mediate mitochondrial fusion in some contexts.

Affected adipose tissue shows strong and consistent evidence of activation of mitochondrial stress responses, with notably strong upregulation of genes known to be positively regulated by ATF4 and CHOP (e.g. TRB3, ATF5, PSAT1, CHAC1, HSPA5 (BiP), SLC7A5, SLC7A11) (*Han et al., 2013*) or mTORC1 signaling. ATF4 and CHOP activities and expression are upregulated by PERK, which mediates a key arm of the ER unfolded protein response, and has also been shown to interact directly with MFN2 at ER-mitochondrial junctions (*Naon and Scorrano, 2014*; *Pablo Muñoz et al., 2014*), while ATF5 has recently been suggested to be the mammalian counterpart of the *C. elegans* mitochondrial unfolded protein sensor ATFS-1 (*Fiorese et al., 2016*). mTORC1 may be activated either downstream or upstream from the UPR (*Appenzeller-Herzog and Hall, 2012*), driving protein synthesis, especially of nucleus-encoded mitochondrial genes (*Morita et al., 2013*).

Increased cell proliferation or survival in the face of mitochondrial dysfunction is not unprecedented: The phenomenon of mitohormesis (*Yun and Finkel, 2014*), whereby mild mitochondrial dysfunction exerts pro-survival effects on cells and organisms, is well established in model systems. MSL is also associated with other forms of genetic or acquired mitochondrial insult, including a common mitochondrial lysyl tRNA mutation (*Chong et al., 2003*), HIV infection treated with highly active anti-retroviral therapy (*Giralt et al., 2011*), and excess alcohol intake (*Enzi et al., 2015*). This raises the possibility that in MFN2-related MSL mitochondrial dysfunction falls in a narrow window of severity that drives mitohormetic upper body preadipocyte differentiation and/or reduced adipocyte clearance. Whether lower body adipocytes are relatively less unaffected, and thus depleted of triglyceride stores as a 'bystander' effect of pathologically expanding upper body adiposity, or whether they are more severely affected and thus die in the face of the same genetic insult, remains to be determined.

Arguing against a critical role for a general reduction in mitochondrial function as the critical driver of MSL, the overwhelming majority of known forms of monogenic mitochondrial dysfunction, of varying severity, are not associated with it, suggesting an alternative, or additional, specific effect of the MFN2 mutations observed. This is plausible as MFN2 deficiency has been shown in cellular contexts to suppress apoptosis (*Pablo Muñoz et al., 2014*; *Shen et al., 2007*), and in ischemic hearts and kidneys to reduce ischaemic tissue (*Hall et al., 2016*; *Pablo Muñoz et al., 2014*) damage and cell death. MFN2-related MSL could thus reflect a dyscoordinated cellular stress response to severe mitochondrial dysfunction, with MFN2 mutations preventing cell death, allowing survival and proliferation of functionally impaired cells.

Finally, our findings have potentially important implications for therapy. Hypogonadism and hyperphagia related to absolute leptin deficiency in P1 may respond to subcutaneous leptin therapy, reducing the burden imposed on overgrown adipose tissue by chronic positive energy balance, while the effect of inhibiting the upregulated anabolic pathways detected transcriptomically, such as mTOR signalling, using clinically available inhibitors, is worthy of consideration. This is in keeping with the promise shown by sirolimus in murine mitochondrial disease (*Johnson et al., 2013*).

## Materials and methods

### Volunteer recruitment

Written informed consent was obtained from all participants or their parents if under 18 years old. The research was approved by relevant UK, US, or Danish Research Ethics Committees and was conducted in accordance with the Declaration of Helsinki.

### Physiological studies and imaging

Biochemical evaluations were performed in accredited diagnostic laboratories. Age- and BMI-matched reference ranges for adiponectin and leptin were derived from the 5-95th percentiles of the MRC Ely study (*Williams et al., 1995*). Oral glucose tolerance tests (OGTTs) were performed with 75 g glucose following an overnight fast. IVGTT was performed as previously described (*Sumner et al., 2004*). Body composition was assessed using Lunar Prodigy dual-energy X-ray absorptiometry (DEXA, GE Lunar). Hepatic triglyceride was assessed in P1 using proton magnetic resonance spectroscopy on a Siemens 3T Verio scanner using methods previously described (*Semple et al., 2009*), and quantified as the ratio of methylene to combined methylene and water signals corrected for spin-spin relaxation. Adipose distribution was determined by magnetic resonance imaging, using T1-weighted turbo spin echo, coronal images, also generated by a Siemens 3T Verio scanner.

### Genetic studies

Exome sequencing of lymphocyte DNA was undertaken in separate laboratories for families of P1-3, in both cases using SureSelect capture (Agilent) followed by HiSeq2000 sequencing (Illumina). Exome-wide sequencing, sequence alignment and variant calling and filtering are described separately below for P1 and P2,3. Sanger sequencing was used for primary genetic diagnosis in P4, and for confirmation of *MFN2* mutations in P1-3.

### Exome-wide sequencing and analysis for P1

Lymphocyte DNA from P1 and her parents was used for exome- sequencing, employing SureSelect capture (Agilent) followed by HiSeq2000 sequencing (Illumina). Sequence data were aligned to the human genome assembly hg19 using the software bwa23(version 0.7.5a-r405) (*Li and Durbin, 2009*) with default parameters. Duplicates were removed, and the mate-pair information was defined using picard tools (http://picard.sourceforge.net) (version 1.95). Subsequently, local realignment surrounding known indel regions was performed, and the quality score for each read was recalibrated with gatk24 (version 2.4–7-g5e89f01) (*McKenna et al., 2010*) using the same resource files as in the 1000 Genomes Project (*Abecasis et al., 2010*). Finally genotypes were called using SAMtools25 (version 0.1.19–44428 cd) http://samtools.sourceforge.net/; SAMTOOLS, RRID:SCR_002105) (*Li, 2011*) and bcftools with standard parameters (-A-c -e -g-v). Variants were annotated with Ensembl Variant Effect Predictor against Ensembl release 84 (*McLaren et al., 2010*) and the NCBI dbSNP database build 144 (ftp://ftp.ncbi.nlm.nih.gov/snp/organisms/human_9606/). The resulting mutation list was interrogated to look for rare sequence variants in *MFN2* based on the strong pretest probability of a mutation based on the published association of MSL, neuropathy and *MFN2* mutations.

### Exome-wide sequencing and analysis for P2,3 and parents

Lymphocyte DNA from P2, P3 and their parents was isolated using the QIAamp and the Autopure kits. Exome Sequencing was undertaken at the Genomic Analysis Facility at the Duke Centre for Human Genome Variation, USA. Exons were captured using the Agilent exon capture platform and sequencing undertaken on the Illumina HiSeq 2000 system. Reads were aligned to the reference sequence using Novoalign alignment tool (http://www.novocraft.com/products/novoalign/; NovoA-lign, RRID:SCR_014818). Depth of capture and completeness of coverage of the exome was assessed using analytical tools built around the BED tools package. Quality filtering was undertaken using the Sam tools and Picard tools packages. Single nucleotide substitutions and small indels were called using the Sam tools package (http://samtools.sourceforge.net/; SAMTOOLS, RRID:SCR_002105) (*Li et al., 2009*). All identified variants were annotated with respect to open reading frames and cross referenced with publically available variant databases and internal control samples using

scripts built around the Annovar tool (http://wannovar.usc.edu/index.php; wANNOVAR, RRID:SCR_000565) (*Yang and Wang, 2015*). Filtering for novel variants was performed by comparison to dbSNP132 and 1000 Genomes SNP calls and patient variants were compared to variants identified in 250 control exomes sequenced and analyzed in a similar manner. 30 rare variants (in coding sequence or splice sites) with an allele frequency <0.01 in control populations (1000 genome and exome variant server) were identified. Of these only the *MFN2* variant was recorded in ClinVar (Clin-Var, RRID:SCR_006169) where is was classified as Pathogenic (https://www.ncbi.nlm.nih.gov/clinvar/RCV000002369/). The Exome Variant Server (http://evs.gs.washington.edu/EVS/) and EXaC Browser (http://exac.broadinstitute.org/gene/ENSG00000116688) datasets were accessed in June 2015.

## Protein structure modelling

The sequences of human MFN2 and Bacterial Dynamin-Like Protein (BDLP) from *Nostoc punctiforme* were aligned using HHpred and PROMALS (*Pei and Grishin, 2007*) algorithms. PDB coordinates of monomer (2J69, [*Low and Löwe, 2006*]), dimer (2W6D, [*Low et al., 2009*]), and tetramer (provided by Dr. Harry Low) were used to display the MFN2 regions of interest in *Figure 1D* and *Figure 1—figure supplement 2*. Model of MFN2 HR2 dimer (*Figure 1E*) and HR1/HR2 heterodimer were built using coordinates of HR2 NFN1 dimer (1T3J, [*Koshiba et al., 2004*]) using Modeller software (MOD-ELLER: Program for Comparative Protein Structure Modelling by Satisfaction of Spatial Restraints, RRID:SCR_008395) (*Sali and Blundell, 1993*).

## Tissue biopsy and processing

Subcutaneous adipose tissue biopsies were taken from the interscapular region under general anaesthesia in the fasting state for P2, P3, and five otherwise healthy volunteers undergoing corrective surgery for idiopathic scoliosis (all female, median age 14.9 (12.9–18.9), median B.M.I. 21.9 kg/m$^2$ (19.0–28.4)). Adipose tissue was dissected from surrounding connective and vascular tissue on dry ice and either frozen in liquid nitrogen for later protein or RNA extraction, fixed in 4% PFA/PBS and paraffin embedded using standard procedures for light microscopy, or fixed for 24 hr in 2% glutaraldehyde/2% formaldehyde in 0.05 M sodium cacodylate at pH 7.4, followed by storage in 0.05 M sodium cacodylate, for electron microscopy. Overlying skin from P1, P2, P3, and two otherwise healthy volunteers biopsies was used to culture dermal fibroblasts using standard procedures. A third dermal fibroblast control cell line was obtained from ATCC (ATCC Cat# CRL-2707, RRID: CVCL_2373).

DNA was extracted using QIAamp DNA Micro Kit (Qiagen). For RNA extraction, three frozen adipose aliquots per subject were transferred into 1 mL QIAzol (Qiagen) and homogenized on thawing with a hand-held TissueRuptor (Qiagen) before centrifugation at 13,000 g for 3 min. Infranatant beneath the fatty layer was mixed with 1 mL 100% ethanol before binding to RNeasy columns and purifying using the RNAeasy kit according to manufacturer's instructions.

Frozen adipose tissue samples were homogenized manually with a pestle in modified RIPA buffer (150 mM NaCl, 1% Nonidet P-40, 0.25% sodium deoxycholate, 50 mM Tris-Cl, pH 7.8) supplemented with Complete Protease Inhibitor cocktail (Roche). Tissue extracts were then cleared by centrifugation. Debris and the top layer of fat were discarded.

Subcutaneous adipose biopsies were also obtained in a separate centre under local anaesthetic from the neck and abdomen of P1. RNA was extracted from these biopsies, and from surgical biopsies obtained from abdominal adipose tissue from ten women undergoing elective abdominal surgery (Median age 68.5 years (range 45–83); median B.M.I. 27.2 kg/m$^2$ (22.2–41.0)).

## mtDNA/nDNA determination

Relative levels of mitochondrial DNA (mtDNA) and nuclear DNA (nDNA) in total DNA were quantified by singleplex quantitative real-time PCR with primers targeting the mitochondria-encoded *ND1* and the nucleus-encoded *B2M* genes followed by evaluation of Ct values using the comparative Ct ($2^{-\Delta\Delta Ct}$) method. Primers and fluorogenic probes used are detailed in *Table 4*. Differences between means were analyzed by two-tailed Student's *t* test using GraphPad Prism 5.0b software (RRID:SCR_002798).

**Table 4.** Primers and Probes used for mRNA and mtDNA quantification.

| Gene | TaqMan GE assay | Taqman GE primers and probe |
|---|---|---|
| PPARGC1A | Hs01016719_m1 | NA |
| PPARGC1B | Hs00991676_m1 | NA |
| NFE2L | Hs00975961_g1 | NA |
| DDIT3 | Hs00358796_g1 | NA |
| HSPA5 | Hs00607129_gH | NA |
| ATF5 | Hs01119208_m1 | NA |
| LEP | Hs00174877_m1 | NA |
| TRIB3 | Hs01082394_m1 | NA |
| PSAT1 | Hs00795278_mH | NA |
| CIDEA | Hs00154455_m1 | NA |
| ADIPOQ | Hs00605917_m1 | NA |
| TBP | Hs99999910_m1 | NA |
| UBC | Hs01871556_s1 | NA |
| YWHAZ | Hs03044281_g1 | NA |
| ND1 | NA | Forward: 5'-CCCTAAAACCCGCCACATCT-3'<br>Reverse: 5'-GAGCGATGGTGAGAGCTAAGGT-3'<br>Probe: [6FAM]—5'-CCATCACCCTCTACATCACCGCCC-3'-[TAM] |
| B2M | NA | Forward: 5'-CCAGCAGAGAATGGAAAGTCAA-3'<br>Reverse: 5'-TCTCTCTCCATTCTTCAGTAAGTCAACT-3'<br>Probe: [6FAM]—5'- ATGTGTCTGGGTTTCATCCATCCGACA-3'-[TAM] |

## Quantitative histomorphometry of paraffin-embedded adipose tissue

Images were captured from sections using an upright Olympus BX-41 microscope (Olympus) equipped with a 20 x objective and a colour ColorView CCD camera (Olympus). A semi-automated workflow in CellP software (Olympus) was used to detect cell membranes of individual adipocytes and determine the cell area. Briefly, images were background subtracted, filtered, thresholded, binarised, prior to ROI inspection and manual correction, cell labelling for traceability, and measurement. GraphPad Prism (RRID:SCR_002798) was used to calculate relative frequency and cumulative distributions. More than 500 adipocytes were measured per condition.

## Gene expression analysis

For RNAseq, RNA was concentrated with the RNeasy MinElute Cleanup Kit (Qiagen) and subject to assessment of quantity and quality using an Agilent BioAnalyzer 2100. RNA samples were whole transcriptome-amplified with NuGEN RNA-Seq System V2 and converted to indexed libraries with NuGEN Ovation Rapid DR Multiplex Library System following the manufacturer's instructions. Libraries were sequenced on an Ilumina HiSeq 4000 system (50 bp, single read sequencing). Cuffdiff from the Samtools suite was used to align reads, FeatureCount to count fragments overlapping annotated human genes, and the resulting data were analysed with the EdgeR package (edgeR, RRID:SCR_012802): normalization used weighted trimmed means of M-values (TMM) and a design matrix comparing affected with unaffected samples. Generalized linear modeling (glmLRT) was used for differential expression analysis. The resulting gene list was analysed using Ingenuity Pathway Analysis (IPA) software. Data were deposited in GEO under accession number GSE97156 (http://www.ncbi.nlm.nih.gov/geo/query/acc.cgi?acc=GSE97156).

Confirmation of selected RNAseq findings was undertaken using TaqMan quantitative real-time PCR using the primer/probesets indicated in *Table 4*. *TBP*, *UBC*, and *YWHAZ* were chosen as endogenous controls based on RNAseq analysis. Serial dilutions of all controls and samples were run, and Ct-values were compared using an efficiency-corrected comparative Ct method based on the geometric mean of the three endogenous controls.

Subcutaneous abdominal RNA and neck adipose RNA from P1 was separately analysed in duplicate in the same batch as subcutaneous abdominal RNA from 10 female controls. RNA was purified using the miRNeasy Mini Kit from Qiagen and mRNA expression measured using the Illumina HT-12 v4 BeadChip. Fold changes in mRNA expression levels were calculated by comparing the median value of P1 with the median value of the 10 female controls.

## Immunoblotting

Proteins were resolved using NuPAGE Novex 4–12% Bis-Tris gels electroblotted onto PVDF membranes using the iBlot system (Invitrogen). Antibodies specific to the following proteins were used: Citrate synthase (Abcam Cat# ab129095 RRID:AB_11143209), MFN1 (Mitofusin-1 (D6E2S, Cell Signaling Techonology), MFN2 (Mitofusin-2 (D2D10), Cell Signaling Technology), Total Human OXPHOS cocktail (Mitosciences, Abcam), γ-tubulin (Sigma-Aldrich Cat# T5326 RRID:AB_532292), TATA binding protein (Abcam Cat# ab818 RRID:AB_306337), β-actin (Sigma-Aldrich Cat# A5316 RRID:AB_476743), PLIN1 (GP33, PROGEN Biotechnik), PLIN2 (GP47, PROGEN Biotechnik), adiponectin (Abcam Cat# ab13881 RRID:AB_2221613).

## Dermal fibroblast studies

Dermal fibroblasts were cultured in DMEM (Invitrogen) supplemented with 10% fetal bovine serum (Hyclone), 1% penicillin-streptomycin and 2 mM L-glutamine (Invitrogen), in an incubator at 37°C in 5% $CO_2$/95% $O_2$ atmosphere. All dermal fibroblast lines were routinely tested as negative for mycoplasma contaminations using VenorGem Classic Mycoplasma Testing PCR Kit (Minerva Biolabs, Cat# 11–1050).

Dermal fibroblast lysates were prepared using M-PER Mammalian Protein Extraction Reagent (Thermo Scientific) supplemented with protease inhibitors. Whole cell extracts were cleared by centrifugation.

For confocal studies of MFN2 localisation, dermal fibroblasts grown on glass coverslips were labeled with MitoTracker Orange CMTMRos dye (Molecular Probes) and then fixed with 4% paraformaldehyde for 15 min at room temperature, permeabilized in 0.05% (v/v) Triton X-100/PBS, and immunostained with antibodies specific to MFN2 in a BSA/PBS blocking solution. Coverslips were mounted in ProLong Gold Antifade Reagent with DAPI mounting medium (Molecular Probes). To inhibit mitochondrial fission dermal fibroblasts were incubated at 37°C with 50 µM mdivi-1 (SIGMA) for 3 hr.

To assess mitochondrial fusion cells stably expressing photoactivatable-GFP (mitoPA-GFP) and matrix-targeted DsRed (mitoDsRed) were first generated. Lentivector particles were produced by transfecting HEK293T cells with either lentiviral mitochondria matrix-targeted PA-GFPmt (a gift from Orian Shirihai, Addgene plasmid # 19989) or lentiviral mitoDsRed (a gift from Pantelis Tsoulfas, Addgene plasmid # 44386), and the lentivirus packaging vectors psPAX2 and pMD2.G using Cal-Phos Mammalian Transfection Kit (Takara) following the manufacturer's guidelines. Dermal fibroblasts were simultaneously transduced with both lentivector particles in the presence of 8 µg/mL polybrene (SIGMA). Resulting cells expressing mitoDsRed and mitoPA-GFP were plated in glass-bottomed microwell dishes (Matek) and imaged live on a SP8 confocal microscope (Leica) equipped with a cage incubator for temperature and $CO_2$/$O_2$ control (Life Imaging Services). PA-GFP was photoactivated in regions of interest within mito-DsRed-expressing mitochondria with a 405 nm laser. The activated fluorescent signal was then collected in confocal image stacks every 5 min for 30 min. Average fluorescence intensities of mitoPA-GFP at each time point was measured in the region of photoconversion, normalized to the mito-DsRed signal, and plotted against time. Images were analysed with the Fiji software package (https://fiji.sc; Fiji, RRID:SCR_002285).

To obtain transmission electron micrographs of mitochondria in dermal fibroblasts, cells were grown on Thermanox coverslips (Nunc) and fixed in 2% glutaraldehyde in 0.1 M phosphate buffer, pH7.4. Cells were then scraped, harvested by centrifugation, and post-fixed in a solution of 1% osmium tetroxide and 1% potassium hexacyanoferrate (II), dehydrated in acetone and finally epoxiresin embedded. Semi-thin sections were stained with toluidine blue. Thin sections were mounted on copper grids, stained with lead citrate, and examined using a Tecnai G2 80-200kv transmission electron microscope.

## Acknowledgements

We are grateful to the patients for their willingness to participate in these studies, to Gregory Strachan of the MRL Imaging Facility (Wellcome Trust Strategic Award [100574/Z/12/Z]) for assistance with imaging and image analysis, to Professor Ram Weiss and Dr Jian'an Luan for providing NEFA data from control groups used in *Figure 1*, to Dr. Harry Low for helpful comments on the structural consequences of the MFN2 mutations and for providing the coordinates of the heterodimer, to Dr. James McMillan at the Cambridge Advanced Imaging Centre, to Dr. Steen Pedersen for obtaining WAT biopsies from P1, and to Adam Neidert, Rita Hench, Diana Rus, and Rasimcan Meral for assistance with P4 studies.

## Additional information

### Competing interests

EAO: Receives grant and consultant fees from Aegerion Pharmaceuticals, and Akcea Therapeutics, scientific advisory board member for AstraZeneca, nonmaterial support from Aegerion Pharmaceuticals and Boehringer Ingelheim. None of these are pertinent for the work here. The other authors declare that no competing interests exist.

### Funding

| Funder | Grant reference number | Author |
|---|---|---|
| Medical Research Council | MRC_MC_UU_12012/5 | Nuno Rocha |
| National Institute for Health Research | Cambridge BRC and Clinical Research Facility; Rare Disease Translational Research Collaboration | Stephen O'Rahilly Eamonn R Maher David B Savage Robert K Semple |
| National Institutes of Health | Oxford-Cambridge Scholars Programme | David A Bulger |
| Wellcome | WT098051 | Felicity Payne Inês Barroso |
| National Institute of Diabetes and Digestive and Kidney Diseases | Intramural research program | Elif A Oral |
| National Institute of Diabetes and Digestive and Kidney Diseases | RO1-DK 08811 | Elif A Oral |
| UCB Pharma | Gen 001 | Hannah Titheradge Duncan McHale Eamonn R Maher |
| Medical Research Council | MRC Centre grant (G0601943) | Mary Reilly |
| National Institutes of Neurological Diseases and Stroke and office of Rare Diseases | U54NS065712 | Mary Reilly |
| Wellcome | Wellcome Trust Postdoctoral Fellowship for Clinicians (10043/Z/15/Z) | Alex Rossor |
| National Institute for Health Research | NIHR Clinical Research Facilities for Experimental Medicine Award to Cambridge NIHR/Wellcome Trust | Alison Sleigh |

The funders had no role in study design, data collection and interpretation, or the decision to submit the work for publication.

## Author contributions
NR, Conceptualization, Data curation, Formal analysis, Investigation, Methodology, Writing—original draft, Writing—review and editing; DAB, Formal analysis, Investigation, Writing—original draft, Writing—review and editing; AF, SBG, Formal analysis, Investigation, Writing—review and editing; HT, Conceptualization, Formal analysis, Investigation, Writing—review and editing; RK, Resources, Investigation, Writing—review and editing, final approval of the version to be published; MP, FP, Formal analysis, Writing—review and editing, final approval of the version to be published; JH, CA, Resources, Investigation, final approval of the version to be published; AS, Formal analysis, Investigation, final approval of the version to be published; JC, Resources, Writing—review and editing, final approval of the version to be published; APG, JB-J, Formal analysis, Investigation, Writing—review and editing, final approval of the version to be published; OP, TH, Supervision, Funding acquisition, Writing—review and editing, final approval of the version to be published; IB, Formal analysis, Supervision, Writing—review and editing, final approval of the version to be published; HC, Resources, Writing—review and editing, Acquired data; MR, AR, Resources, Writing—review and editing, Final approval of the version to be published; RJB, SIT, Writing—review and editing, Acquired data; final approval of the version to be published; DM, Supervision, Funding acquisition, Writing—review and editing, Final approval of the version to be published; MA, Formal analysis, Writing—review and editing; EAO, Data curation, Writing—original draft, Writing—review and editing; VS, Formal analysis, Writing—original draft, Writing—review and editing; SO'R, Supervision, Funding acquisition, Writing—review and editing; ERM, BR, Conceptualization, Supervision, Investigation, Writing—original draft, Writing—review and editing; DBS, Conceptualization, Formal analysis, Funding acquisition, Investigation, Writing—original draft, Writing—review and editing; RKS, Conceptualization, Formal analysis, Supervision, Funding acquisition, Investigation, Writing—original draft, Writing—review and editing

## Author ORCIDs
Inês Barroso, http://orcid.org/0000-0001-5800-4520
Robert K Semple, http://orcid.org/0000-0001-6539-3069

## Ethics
Human subjects: Written informed consent was obtained from all participants or their parents if under 18 years old for the research described and for publication of results. The research was approved by the Cambridgeshire South Research Ethics Committee, Reference no. 12/EE/0405

# Additional files

## Major datasets
The following dataset was generated:

| Author(s) | Year | Dataset title | Dataset URL | Database, license, and accessibility information |
|---|---|---|---|---|
| Bulger DA, Lam B | 2017 | RNA-seq of overgrown interscapular adipose tissue with biallelic MFN2 mutations | http://www.ncbi.nlm.nih.gov/geo/query/acc.cgi?acc=GSE97156 | Publicly available at the NCBI Gene Expression Omnibus (accession no: GSE97156) |

The following previously published datasets were used:

| Author(s) | Year | Dataset title | Dataset URL | Database, license, and accessibility information |
|---|---|---|---|---|
| Landrum MJ, Lee JM, Benson M, Brown G, Chao C, Chitipiralla S, Gu B, Hart J, Hoffman D, | 2016 | ClinVar: public archive of interpretations of clinically relevant variants | https://www.ncbi.nlm.nih.gov/clinvar/RCV000002369/ | Publicly available at the NCBI ClinVar (accession no: RCV000002369) |

| | | | | |
|---|---|---|---|---|
| Hoover J, Jang W, Katz K, Ovetsky M, Riley G, Sethi A, Tully R, Villamarin-Salomon R, Rubinstein W, Maglott DR | | | | |
| Lek M, Karczewski KJ, Minikel EV, Samocha KE, Banks E, Fennell T, O'Donnell-Luria AH, Ware JS, Hill AJ, Cummings BB, Tukiainen T, Birnbaum DP, Kosmicki JA, Duncan LE, Estrada K, Zhao F, Zou J, Pierce-Hoffman E, Berghout J, Cooper DN, Deflaux N, DePristo M, Do R, Flannick J, Fromer M, Gauthier L, Goldstein J, Gupta N, Howrigan D, Kiezun A, Kurki MI, Moonshine AL, Natarajan P, Orozco L, Peloso GM, Poplin R, Rivas MA, Ruano-Rubio V, Rose SA, Ruderfer DM, Shakir K, Stenson PD, Stevens C, Thomas BP, Tiao G, Tusie-Luna MT, Weisburd B, Won HH Yu D Altshuler DM, Ardissino D, Boehnke M, Danesh J, Donnelly S, Elosua R, Florez JC, Gabriel SB, Getz G, Glatt SJ Hultman CM Kathiresan S, Laakso M McCarroll S, McCarthy MI, McGovern D, McPherson R, Neale BM, Palotie A, Purcell SM, Saleheen D Scharf JM Sklar P Sullivan PF, Tuomilehto J, Tsuang MT, Watkins HC, Wilson JG, Daly MJ, MacArthur DG, Exome Aggregation Consortium | 2016 | Exome Aggregation Consortium (ExAC) | http://exac.broadinstitute.org/gene/ENSG00000116688 | Publicly available at Broad Institute (accession no: ENSG00000116688) |
| Seattle WA | 2015 | Exome Variant Server | http://evs.gs.washington.edu/evs_bulk_data/ESP6500SI-V2-SSA137.GRCh38-liftover.snps_indels.txt.tar.gz | Publicly available via the Exome Variant Server |
| Cunningham F, Amode MR, Barrell D, Beal K, Billis K, Brent S, Carvalho-Silva D, Clapham P, Coates G, Fitzgerald S, Gil L, Girón | 2015 | Ensembl 2015 | ftp://ftp.ensembl.org/pub/current_embl/homo_sapiens/ | Publicly available at Ensembl |

CG, Gordon L, Hourlier T, Hunt SE, Janacek SH, Johnson N, Juettemann T, Kähäri AK, Keenan S, Martin FJ, Maurel T, McLaren W, Murphy DN, Nag R, Overduin B, Parker A, Patricio M, Perry E, Pignatelli M, Riat HS, Sheppard D, Taylor K, Thormann A, Vullo A, Wilder SP, Zadissa A, Aken BL, Birney E, Harrow J, Kinsella R, Muffato M, Ruffier M, Searle SM, Spudich G, Trevanion SJ, Yates A, Zerbino DR, Flicek P

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
