## [Decision Letter]

Thank you for submitting your article "Human biallelic MFN2 mutations induce mitochondrial dysfunction, adipose hyperplasia, and suppressed leptin expression" for consideration by *eLife*. Your article has been reviewed by two peer reviewers, and the evaluation has been overseen by a Reviewing Editor and Mark McCarthy as the Senior Editor. The following individual involved in review of your submission has agreed to reveal his identity: Fredrik Karpe (Reviewer #3).

The reviewers have discussed the reviews with one another and the Reviewing Editor has drafted this decision to help you prepare a revised submission.

Summary:

Both reviewers consider the work performed impressive and comprehensive. The strength and novelty of the report is the detailed structural studies of adipocytes (EM) demonstrating a severe disruption of mitochondrial integrity and the *in vivo* metabolic studies of the patients. However, the reviewers and editors have additional suggestions, that seek to more clearly separate what was already known, to put new observations into context, and to further strengthen the mechanistic inference.

We summarize the reviews, which will require you to make textual changes and (potentially) include additional experimental data, (subject to availability of relevant samples).

Revisions:

A) The Introduction needs further details on what is already know about MSL (and commensurately less broad discussion regarding adipose tissue and disease).

B) We would like to see more detailed discussion of the current findings in light of what was already known, emphasizing what is novel and what was already established before.

C) We are keen to deepen the mechanistic explanation of the consequences of the causing mutations, ideally through responses on the following issues raised by the reviewers:

1) The FFA response in the most affect case is counterintuitive. How can the tissue expand to such proportions if lipolysis is not controlled (down-regulated by insulin)? Unopposed lipolysis would rapidly deplete adipocytes of their triacylglycerol content in the affected depots and one would need to either postulate a very active uptake mechanism of FFA with an even more active re-esterification of fatty acids balancing the constant outward flux of FFA. An alternative hypothesis is that adipose tissues that seem depleted (leg and arm) are the origin of the excess FFA, but structural studies of those tissues have not been made. The RNASeq data could be used to interrogate uptake pathways and storage pathways. A fatty acid profile of the tissue would be of interest. Although rather unlikely as an explanation, it could reveal unusual input of fatty acids, such as from de novo lipogenesis.

2) Ideally, affected and unaffected adipose tissue from the same individual should have been studied, but it is appreciated this could have been logistically difficult. It would be helpful if the mitochondria of the cultured skin fibroblasts could be studied by EM to see if the aberrant structures are identified. If relevant tissue is available, we would like to see these experiments done as part of the revision.

3) More explicit description of how the data supports the inference of mitochondrial dysfunction; gene expression by itself is not sufficient to claim mitochondrial functional defect

4) It would be interesting if the investigators could comment more on the fat depots that seem depleted. Are they depleted because there is a 'steal' phenomenon by the aberrantly expanded tissue (in line with the mosaic overgrowth due to PIKC3 (see Figure 1) or is a question of global effects on adipose tissues where certain tissue are affected and not other (such as in Dunnigan Kobberling) by as yet unexplained mechanisms?

---

## [Author Response]

*We summarize the reviews, which will require you to make textual changes and (potentially) include additional experimental data, (subject to availability of relevant samples).*

*Revisions:*

*A) The Introduction needs further details on what is already know about MSL (and commensurately less broad discussion regarding adipose tissue and disease).*

Although MSL is a spectacular phenotype, the pertinent literature is somewhat fragmented, being largely devoted to clinical phenomenology. We don’t believe there to be important aspects that we have not touched upon in the Introduction. Nevertheless in an attempt to focus better we have edited both by slimming down the general preamble and by adding selected further observations. Two reviews referenced in the text (Herbst, 2012 and Enzi et al., 2015) and published in the last 5 years summarise much of the past literature – we have referred to both of these extensively. Importantly, the past literature is very varied (as highlighted by Herbst, 2012), especially in relation to key aspects of the phenotype we report including leptin deficiency and perturbations in lipolysis. We suspect that this relates to heterogeneity in underlying pathogenetic mechanisms – we have tried to make this point more clearly in the revised manuscript. We shall not reproduce the whole introductory section here.

*B) We would like to see more detailed discussion of the current findings in light of what was already known, emphasizing what is novel and what was already established before.*

Again, as noted above, studies of MSL over many decades have been scattered and predominantly descriptive, focussing on clinical phenomenology. Moreover the condition is somewhat heterogeneous, complicating interpretation of the limited number of cellular and other studies that have been undertaken. We have cited such evidence that exists, including recent good reviews, but have spent more time putting this work into the context of the large bodies of detailed mechanistic studies of mitofusin 2 *in vivo* and in cell models, and wider studies of mitochondrial function and adipocytes. We believe that among the most important aspects of this study is the directing of the sophisticated field of mitofusin biology towards this human entity which has not been explained or anticipated by any prior cellular or mouse studies. We are conscious that the discussion is long in order to accommodate the many important bodies of work of relevance to our findings, but we are reluctant not to make comment on these. We have slightly rearranged discussion and have added an introductory paragraph to make clear what is new, however if the reviewers and editor take a different view of what is required, we should be happy to edit further according to their guidance.

*C) We are keen to deepen the mechanistic explanation of the consequences of the causing mutations, ideally through responses on the following issues raised by the reviewers:*

*1) The FFA response in the most affect case is counterintuitive. How can the tissue expand to such proportions if lipolysis is not controlled (down-regulated by insulin)? Unopposed lipolysis would rapidly deplete adipocytes of their triacylglycerol content in the affected depots and one would need to either postulate a very active uptake mechanism of FFA with an even more active re-esterification of fatty acids balancing the constant outward flux of FFA. An alternative hypothesis is that adipose tissues that seem depleted (leg and arm) are the origin of the excess FFA, but structural studies of those tissues have not been made. The RNASeq data could be used to interrogate uptake pathways and storage pathways. A fatty acid profile of the tissue would be of interest. Although rather unlikely as an explanation, it could reveal unusual input of fatty acids, such as from* de novo *lipogenesis.*

Whilst we agree that truly “unopposed lipolysis” would be expected to deplete TAG content in affected adipocytes, this is not how we interpret the observations reported and is not what we meant to suggest. We have tempered our description of lipolytic suppression to reflect our interpretation of the observations and the potential involvement of the lower limb adipose depots more accurately. The data we have derives from 2 OGTTs in proband 1, 1 OGTT in each of Probands 2 and 3, and now a frequently sampled IVGTT in (new) proband 4. Proband 1’s data showed reproducible and striking impairment of NEFA suppression during the OGTT, a pattern that was recapitulated, also rather strikingly, in the IVGTT of Proband 4. Fasting NEFA levels were unremarkable (though the reference range is wide). We maintain that this is a very interesting and potentially important observation, however the difference we describe relates only to the perturbation studied (elevated glucose and insulin) and does not equate to “unopposed lipolysis”. We also note that suppression of lipolysis is often reported to be at least slightly delayed in common forms of obesity, so the apparent discordance between increased adiposity and an increase in a surrogate marker of lipolysis under these conditions is not unprecedented – we conclude that it is possible to counterbalance this degree of lipolytic dysregulation.

As suggested we did review the gene expression data we have using a variety of approaches ranging from Ingenuity Pathway Analysis (IPA), KEGG Pathway Analysis, and interrogation of manually curated pertinent gene sets. In short, while there was broad upregulation of many anabolic pathways (e.g. lipid and protein synthesis), the profile of fatty acid transporters and genes involved in FFA esterification and intracellular binding was indeterminate, with no concerted direction of change in expression. This evidence does not definitively rule out high FFA influx, not least as these pathways are subject to many posttranscriptional regulatory steps. We are committed to making all gene expression data and analytic scripts publicly available so that readers can review this type of issue themselves.

The reviewer’s suggestion that the ‘lipodystrophic’ or ‘depleted’ adipose depots might be the source of altered lipolytic regulation is an interesting idea which we have also considered – however, without access to biopsy material from femorogluteal adipose tissue before it is depleted, and with no opportunity so far to undertake depot-selective flux studies, we cannot address this experimentally at present. Nevertheless, we have revised the text to mention this possibility explicitly.

Finally, we do not unfortunately have residual adipose tissue available with which to undertake fatty acid profiling at this stage.

*2) Ideally, affected and unaffected adipose tissue from the same individual should have been studied, but it is appreciated this could have been logistically difficult. It would be helpful if the mitochondria of the cultured skin fibroblasts could be studied by EM to see if the aberrant structures are identified. If relevant tissue is available, we would like to see these experiments done as part of the revision.*

We strongly agree that it would be highly informative to compare adipose tissue from the legs, where lipodystrophy develops, with adipose tissue from an overgrown upper body depot. The central problem with achieving this is that a tissue that is not present (as in patients 1 and 4) cannot be biopsied, and if a “blind” biopsy were taken of subcutaneous tissue from the gluteofemoral region, gene expression would likely be dominated by whatever residue of interstitial cell types remained. Gene expression of adipose tissue from the neck and abdomen of P1 was undertaken using Affy microarray, as already noted, however adipose depots were still present and overgrown to different degrees at both sites, and the gene expression proved to be highly correlated, with analysis of either dataset compared to healthy control, adipose tissue yielding essentially superimposable results:

Author response image 1.Affymetrix Array-determined gene expression in subcutaneous abdominal adipose biopsy (Y axis) compared to adipose tissue derived from the neck (X axis) of P1.**DOI:**
http://dx.doi.org/10.7554/eLife.23813.024

Biopsy from P2 and P3, who still have some adipose tissue in the legs, would have been very attractive, but consent was not given for this.

We have, however, had the opportunity to examine dermal fibroblasts from P1, 2, and 3 ultrastructurally using transmission electron microscopy, as requested, and have shown mitochondria in the cells to have normal morphology. Representative images have now been added as Figure 4—figure supplement 3, and are now commented on in results as follows:

“Furthermore, immunocytochemistry showed appropriate localisation of MFN2 expression to the mitochondrial network (Figure 4, Figure 4—figure supplement 1), which in patient cells appeared morphologically indistinguishable from control cells both at baseline (Figure 4), and after treatment with the small molecule mitochondrial fission inhibitor mdivi-1 (Figure 4—figure supplement 2). Transmission electron microscopy of dermal fibroblasts from P1-3 revealed normal mitochondrial ultrastructure, with no evidence of the striking fragmentation and dysmorphism seen in adipose biopsies (Figure 4—figure supplement 3). Diffusion of photoactivated GFP within the network was also normal when assessed in living cells (Figure 4), although this diffusion was impaired in wild type cells by hydrogen peroxide treatment to fragment the network (data not shown). Mitochondrial membrane potential was unchanged in patient cells, as assessed by a mitochondrial membrane potential sensor JC-1 ratio assay (Figure 4).”

We further note the observation that TEM of the liver from new patient 4 was undertaken at the NIH as part of her workup and the mitochondria were specifically commented on as showing normal morphology. At the time of writing we have not yet been able to secure the primary images, however they were reported by an experienced pathologist specifically in the light of the high lactate level exhibited by the patient.

*3) More explicit description of how the data supports the inference of mitochondrial dysfunction; gene expression by itself is not sufficient to claim mitochondrial functional defect*

In the original manuscript we presented several different lines of evidence for mitochondrial dysfunction (including elevated lactate, abnormal mitochondrial protein expression, and abnormal mitochondrial morphology as well as corresponding transcriptomic profile), and we summarised these explicitly in a dedicated paragraph of the discussion. We have now attempted to give this summary further prominence in a new introductory paragraph to the Discussion, while retaining most of the prior paragraph as below:

“Evidence of impaired mitochondrial function in MFN2-related MSL is seen in strikingly elevated blood lactate concentrations in P1, P4 and previously described patients (Sawyer et al., 2015), reduced expression of citrate synthase and respiratory chain components, and highly abnormal mitochondrial ultrastructure in affected adipose depots. The reduction in mitochondria-encoded transcripts, contrasting with increased nucleus-encoded mitochondrial transcripts and mitochondrial biogenesis factors suggests that compensatory transcriptional programmes are activated to try to restore cellular mitochondrial function. In contrast, gene expression and mitochondrial morphology and fusion of primary dermal fibroblasts is normal, confirming mitochondrial dysfunction in MFN2-related MSL to be tissue-selective, and suggesting that MFN2 Arg707Trp remains able to mediate mitochondrial fusion in some contexts.”

*4) It would be interesting if the investigators could comment more on the fat depots that seem depleted. Are they depleted because there is a 'steal' phenomenon by the aberrantly expanded tissue (in line with the mosaic overgrowth due to PIKC3 (see Figure 1) or is a question of global effects on adipose tissues where certain tissue are affected and not other (such as in Dunnigan Kobberling) by as yet unexplained mechanisms?*

We agree that this is a fascinating aspect of the phenotype, however the dramatic differences between areas of adipose overgrowth compared with lower limb ‘lipodystrophy’ observed most strikingly in proband 1 and now in (new) proband 4 cannot be explained at this stage given considerations above. The terminology “affected” and “unaffected” should not be applied to different adipose depots in MFN2-related MSL until this is clarified; it is possible that femorogluteal fat is more affected than upper body fat, so that adipocytes die due to mitochondrial dysfunction, causing lipodystrophy, while the same insult is more tolerated in the upper body, where a sublethal mitochondrial insult is translated into a hyperplasic response, in accordance with the notion of mitohormesis.

We note that depot-selective variation in adipose tissue is seen in several types of partial lipodystrophy, most prominently in the commonest monogenic form, LMNA-associated FPLD2. In this condition neck and facial fat are selectively preserved, while in PPARG-associated FPLD3 where abdominal fat seems to be preserved in many cases. In these instances, although the clinical syndromes have been known for nearly 20 years, mechanistic understanding also remains elusive. Further understanding of the issue in MFN2-related MSL will likely require identification of additional patients willing to undertake detailed physiological studies involving stable isotope flux analyses in ‘overgrown’ and if possible ‘lipodystrophic’ regions, and or biopsy of paired upper body and femorogluteal adipose tissue while this still exists (likely in children).